# Activity-dependent switch of GABAergic inhibition into glutamatergic excitation in astrocyte-neuron networks

Gertrudis Perea[1]*[†], Ricardo Gómez[1,2][†], Sara Mederos[1], Ana Covelo[3], Jesús J Ballesteros[4,5], Laura Schlosser[6], Alicia Hernández-Vivanco[1], Mario Martín-Fernández[3], Ruth Quintana[3], Abdelrahman Rayan[5], Adolfo Díez[3], Marco Fuenzalida[7], Amit Agarwal[8], Dwight E Bergles[8], Bernhard Bettler[9], Denise Manahan-Vaughan[5], Eduardo D Martín[4], Frank Kirchhoff[6], Alfonso Araque[3]*

[1]Consejo Superior de Investigaciones Científicas, Instituto Cajal, Madrid, Spain; [2]Cellular and Systems Neurobiology, Systems Biology Program, Centre for Genomic Regulation, The Barcelona Institute of Science and Technology, Barcelona, Spain; [3]Department of Neuroscience, University of Minnesota, Minneapolis, United States; [4]Albacete Science and Technology Park, Institute for Research in Neurological Disabilities, University of Castilla-La Mancha, Albacete, Spain; [5]Department of Neurophysiology, Faculty of Medicine, Ruhr University Bochum, Bochum, Germany; [6]Molecular Physiology, Center for Integrative Physiology and Molecular Medicine, University of Saarland, Homburg, Germany; [7]Center of Neurobiology and Brain Plasticity, Institute of Physiology, Faculty of Science, Universidad de Valparaíso, Valparaiso, Chile; [8]Department of Neuroscience, Johns Hopkins School of Medicine, Baltimore, United States; [9]Department of Biomedicine, University of Basel, Basel, Switzerland

*For correspondence: gperea@cajal.csic.es (GP); araque@umn.edu (AAr)

[†]These authors contributed equally to this work

Competing interests: The authors declare that no competing interests exist.

**Abstract** Interneurons are critical for proper neural network function and can activate $Ca^{2+}$ signaling in astrocytes. However, the impact of the interneuron-astrocyte signaling into neuronal network operation remains unknown. Using the simplest hippocampal Astrocyte-Neuron network, i.e., GABAergic interneuron, pyramidal neuron, single CA3-CA1 glutamatergic synapse, and astrocytes, we found that interneuron-astrocyte signaling dynamically affected excitatory neurotransmission in an activity- and time-dependent manner, and determined the sign (inhibition *vs* potentiation) of the GABA-mediated effects. While synaptic inhibition was mediated by $GABA_A$ receptors, potentiation involved astrocyte $GABA_B$ receptors, astrocytic glutamate release, and presynaptic metabotropic glutamate receptors. Using conditional astrocyte-specific $GABA_B$ receptor (*Gabbr1*) knockout mice, we confirmed the glial source of the interneuron-induced potentiation, and demonstrated the involvement of astrocytes in hippocampal theta and gamma oscillations in vivo. Therefore, astrocytes decode interneuron activity and transform inhibitory into excitatory signals, contributing to the emergence of novel network properties resulting from the interneuron-astrocyte interplay.

## Introduction

Interneurons are involved in fundamental aspects of brain function playing a key role in the operation of neuronal networks (*Kullmann, 2011*). GABA (γ-aminobutyric acid)-ergic interneurons control both

the number and the firing frequency of pyramidal cells, synchronize principal cell population discharge contributing to the generation of rhythmic activity in neuronal networks, such as theta and gamma frequency oscillations (*Bartos et al., 2007*; *Kullmann, 2011*; *Pouille and Scanziani, 2001*), and also stimulate astrocyte $Ca^{2+}$ signaling (*Kang et al., 1998*).

Astrocytes, considered for decades to play merely supportive roles for neurons, have emerged as active regulatory elements directly involved in synaptic physiology (*Perea et al., 2009*). Astrocytes sense and integrate synaptic activity by responding with intracellular $Ca^{2+}$ elevations to different neurotransmitters that activate membrane transporters and receptors, and intracellular signaling pathways. Although controversial (*Agulhon et al., 2010*; *Hamilton and Attwell, 2010*), astrocyte $Ca^{2+}$ signal have been shown to stimulate the release of active substances, so-called gliotransmitters, that can regulate neuronal excitability and synaptic transmission and plasticity (*Araque et al., 2014*; *Haydon and Carmignoto, 2006*; *Perea et al., 2009*; *Volterra and Meldolesi, 2005*). Therefore, the functional interaction between astrocytes and neurons suggests an active role of astrocytes in brain function (*Araque et al., 2014*; *Pannasch and Rouach, 2013*).

The existence of signaling between GABAergic interneurons and astrocytes has been demonstrated showing that GABA released from interneurons lead to astrocyte $Ca^{2+}$ elevations mediated by activation of $GABA_B$ receptors (*Andersson et al., 2007*; *Kang et al., 1998*; *Serrano et al., 2006*). These $Ca^{2+}$ elevations may lead to the potentiation of inhibitory synaptic transmission by glutamate released from astrocytes (*Kang et al., 1998*), and the heterosynaptic depression of excitatory transmission by adenosine derived from astrocytic ATP (*Andersson et al., 2007*; *Chen et al., 2013*; *Serrano et al., 2006*). Giving the limited and divergent previous data, the fundamental aspects of the functional relationships between interneurons and astrocytes are still poorly defined, especially regarding their consequences on neuronal network signaling.

Therefore, to investigate the impact and functional consequences of interneuron-astrocyte signaling in neural circuits, we selected a simple archetypical Astrocyte-Neuron network comprising a single CA3-CA1 glutamatergic synapse, a single GABAergic interneuron, a single pyramidal neuron, and astrocytes in the CA1 region of the hippocampus. We found that the precise timing and strength of interneuron activity controls the excitatory synaptic transmission, which is either inhibited by single action potentials (APs) or potentiated by high firing rates. While inhibition was mediated by presynaptic $GABA_A$ receptors, the potentiation required the activation of astrocytic $GABA_B$ receptors, intracellular $Ca^{2+}$ elevations, glutamate release from astrocytes, and activation of presynaptic group I metabotropic glutamate receptors (mGluRs). The dual effect of interneurons was mediated by activity- and time-dependent stimulation of different signals triggered by presynaptic $GABA_A$ and astrocytic $GABA_B$ receptors, which resulted in the dynamic regulation of excitatory transmission. To assess the astrocytic function of $GABA_B$ receptors and their role in excitation boosting, we generated genetically modified mice with conditional ablation of $GABA_B$ receptors in astrocytes (GB1-cKO). For that purpose we crossed GLAST-CreERT2 knockin mice (*Mori et al., 2006*) with *Gabbr*1$^{fl/fl}$ mice (*Haller et al., 2004*). In these mice we observed the absence of excitatory synaptic potentiation by interneuron stimulation. Furthermore, complex circuit activities, such as theta and gamma oscillations, were regulated by GABAergic-astrocytic signaling in vivo, showing astrocyte-specific GB1-cKO mice a notable reduction of those rhythms. In summary, we show that, besides synaptic inhibition, interneuron activity may exert new additional forms of synaptic regulation, which involve the participation of astrocytes, and that the activity-dependent interplay between interneuron-astrocyte-excitatory synapses contributes to the emergence of new dynamic properties of neural network function.

## Results

### Interneurons inhibit or potentiate neurotransmitter release at single CA3-CA1 synapses in an activity-dependent manner

To investigate the consequences of interneuron-astrocyte signaling on excitatory synaptic transmission, we performed whole-cell recordings from pairs of GABAergic interneurons and CA1 pyramidal neurons, while monitoring both the excitatory postsynaptic transmission (EPSC) at putative single CA3-CA1 synapses (*Dobrunz and Stevens, 1997*; *Navarrete and Araque, 2010*; *Perea and Araque, 2007*) and $Ca^{2+}$ signals in astrocytes (*Figure 1A* and *Figure 1—figure supplement 1A–C*).

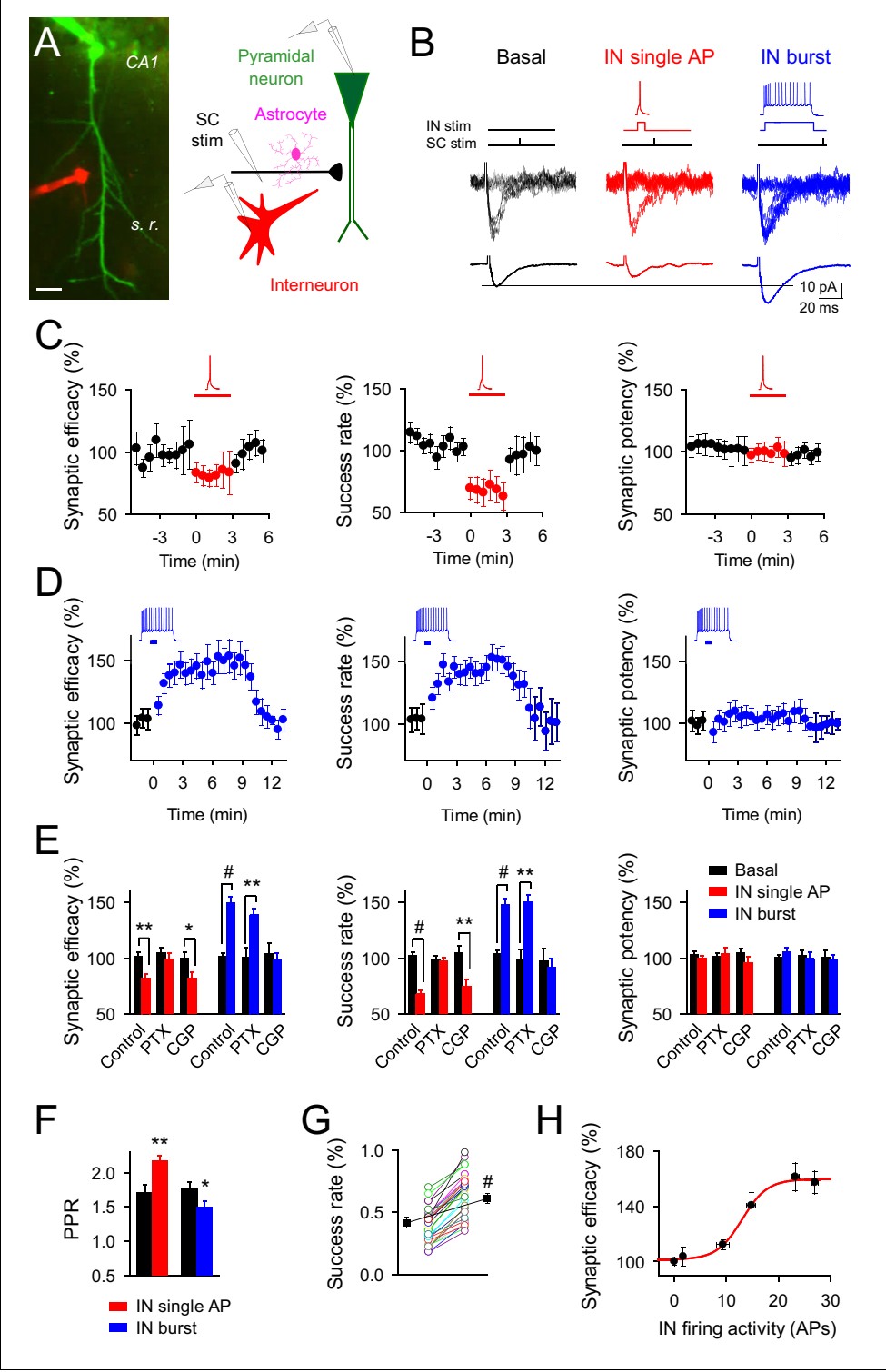

**Figure 1.** Changes on interneuron firing activity inhibit or potentiate transmitter release at single CA3-CA1 synapses. (A) *Left*, image of paired recorded GABAergic interneuron (IN; red) and CA1 pyramidal neuron (green). *Right*, scheme depicting paired recordings from interneuron and pyramidal neurons, and Schaffer collateral (SC) stimulating electrode. Scale bar, 25 mm. (B) Responses evoked by minimal stimulation showing regular EPSC amplitudes and transmission failures (15 consecutive stimuli; top), and averaged EPSCs (50 consecutive stimuli; bottom) before and after pairing SC stimuli with single (red) or bursts (blue) of interneuron APs. (C–D) Synaptic efficacy (i.e., mean amplitude of responses including successes and failures); success rate (percentage of effective

*Figure 1 continued on next page*

*Figure 1 continued*

EPSCs from the total number of stimuli); and synaptic potency (i.e., mean EPSC amplitude excluding failures) (bin width, 33 s) before and after pairing SC stimuli with single (**C**) red; *n* = 10) or bursts (**D**) blue; *n* = 33) of interneuron APs. Horizontal bars indicate the time of pairing. (**E**) Relative changes of synaptic parameters from basal (black) elicited by pairing SC stimuli with single (red) or bursts of interneuron APs (blue), in control (*n* = 10 and 33, for red and blue, respectively), picrotoxin (PTX; *n* = 6 and 7, for red and blue, respectively), and CGP55845 (*n* = 5 and 9, for red and blue, respectively). (**F**) Paired pulse ratio (PPR). (**G**) Success rate from synapses shown in **D**, and averaged values (black) in control and after IN burst stimulation (n = 33). The IN-mediated effects were independent on their initial values. (**H**) Relative potentiation of synaptic efficacy *vs.* interneuron firing activity (*n* $\geq$ 4 for each data point; Hill equation fitting, $R^2$ = 0.9955). *p<0.05, **p<0.01, #p<0.001; paired *t* test. See also *Figure 1—figure supplements 1* and *2*. Error bars indicate SEM, as in all other figures.

The following figure supplements are available for figure 1:

**Figure supplement 1.** Electrophysiological properties of recorded cells and synaptic currents evoked by minimal stimulation.

**Figure supplement 2.** Parvoalbumin-positive CA1 interneuron stimulation potentiates excitatory synaptic transmission at single CA3-CA1 synapses.

We paired interneuron single APs elicited by 15 ms depolarizing pulses with single stimuli at Schaffer collaterals (SC) (pairing: 3 min every 4 s, 10 ms delay) (*Figure 1B*). In 10 out of 48 pairs, interneuron single APs inhibited the CA3-CA1 synaptic efficacy, i.e., the mean EPSC peak amplitude including failures in synaptic transmission (from 9.13 ± 1.44 pA to 6.93 ± 1.10 pA after interneuron stimulation; 82.16 ± 3.52% of basal; *n* = 10; p=0.003; paired *t* test), which resulted from inhibiting the probability of release (success rate; from 0.46 ± 0.05 to 0.29 ± 0.04 after interneuron stimulation; 67.71 ± 3.59% of basal; p<0.001; paired *t* test) without modifying the amplitude of the synaptic potency, i.e., the average of the peak EPSC amplitude when failures are excluded (from 18.37 ± 2.68 pA to 17.33 ± 2.92 pA after interneuron stimulation; 99.38 ± 2.49% of basal; p=0.45; paired *t* test) (*Figure 1C,E*), suggesting a presynaptic mechanism of action. Accordingly, the paired pulse facilitation (PPF) ratio increased after interneuron single action potential stimulation (from 1.70 ± 0.11 to 2.17 ± 0.08 after interneuron stimulation; *n* = 10; p=0.009; paired *t* test) (*Figure 1F*).

We then stimulated the interneuron with longer depolarizing pulses (700 ms) to elicit bursts of APs that were followed by the SC stimulus (*Figure 1B*). In 33 out of 64 pairs, this stimulation paradigm enhanced the synaptic efficacy (from 9.48 ± 0.74 pA to 13.89 ± 1.09 pA after interneuron stimulation; 148.80 ± 5.86% of basal; *n* = 33; p<0.001; paired *t* test) due to an increase of the success rate (from 0.42 ± 0.04 to 0.60 ± 0.04 after interneuron stimulation; 147.62 ± 4.94% of basal; p<0.001; paired *t* test), without affecting the synaptic potency (from 18.79 ± 1.42 pA to 19.78 ± 1.51 pA after interneuron stimulation; 105.40 ± 4.03% of basal; p=0.37; paired *t* test) (*Figure 1D,E*). The increase of success rate and the constancy of synaptic potency indicate a presynaptic mechanism, as supported by the decrease in the PPF ratio (from 1.76 ± 0.10 to 1.48 ± 0.09; *n* = 33; p=0.04; paired *t* test) (*Figure 1F*). The interneuron-mediated synaptic potentiation was observed for synapses showing a wide-range of the success rate (*Figure 1G*; *n* = 33; p<0.001; paired *t* test), indicating that the capability of synaptic modulation was unrelated with their initial values of transmitter release.

Both regulatory phenomena were mediated by GABAergic signaling, but while synaptic inhibition was abolished by picrotoxin (GABA$_A$ receptor antagonist; 50 μM) without affecting the synaptic enhancement (*Figure 1E*), the synaptic potentiation was impaired by CGP55845 (GABA$_B$ receptor antagonist; 5 μM) without modifying inhibition (*Figure 1E*). Therefore, interneuron single APs inhibited CA3-CA1 synapses through activation of presynaptic GABA$_A$ receptors, whereas high interneuron firing rates surprisingly potentiated excitatory synaptic transmission through activation of GABA$_B$ receptors. The degree of synaptic potentiation was activity-dependent, varying with the interneuron number of APs according to the Hill equation (*Figure 1H*), but was independent of the interneuron subtype stimulated (*Figure 1—figure supplement 1D,E*). In order to test the cell-type influence to the interneuron-induced synaptic potentiation, parvoalbumin-positive (PV[+]) interneuron activity was evaluated. Pairs of CA1 pyramidal neuron and PV[+]-interneurons were recorded from PV-

tdTomato transgenic mice (*Figure 1—figure supplement 2*), showing a marked potentiation of excitatory synaptic transmission after PV$^+$-cell burst stimulation (success rate: 146.02 ± 17.86% of basal; n = 9; p=0.002; paired *t* test; *Figure 1—figure supplement 2C,D*), with analogous features to those synapses from unlabeled interneuron recordings (*Figure 1D,E*), supporting that interneuron potentiation of excitatory synaptic transmission might be a broad phenomenon involving different interneuron subtypes.

We next investigated whether inhibition and potentiation of excitatory transmission induced by interneuronal activity were segregated processes to a particular set of synapses or could concur at the same synapse. We found that ~17% of recorded synapses (6 out of 36 recorded pairs) showed both inhibition and potentiation of EPSCs when single and bursts of interneuron APs were consecutively evoked (*Figure 2A,B*). These synapses showed similar properties to those that expressed inhibition and potentiation independently (see *Figures 1C,D* and *2C*), indicating that the mechanisms of synaptic inhibition can coexist with those responsible for synaptic potentiation at the same synapses, and the final outcome is regulated by the interneuron firing rate.

## Interneuron-induced excitatory synaptic potentiation requires astrocyte GABA$_B$ receptor-mediated Ca$^{2+}$ signaling

Because interneuron activity has been shown to elevate astrocyte Ca$^{2+}$ via activation of astrocytic GABA$_B$ receptors (*Kang et al., 1998*; *Mariotti et al., 2016*; *Serrano et al., 2006*), and Ca$^{2+}$-dependent release of gliotransmitters from astrocytes modulate synaptic transmission (*Di Castro et al., 2011*; *Henneberger et al., 2010*; *Jourdain et al., 2007*; *Navarrete et al., 2012*; *Panatier et al., 2011*; *Perea and Araque, 2007*; *Serrano et al., 2006*), we investigated the participation of the astrocyte Ca$^{2+}$ signal in the interneuron-induced synaptic potentiation. We simultaneously recorded from interneurons and monitored Ca$^{2+}$ levels at the soma of the astrocytes (*Figure 3A*). While astrocyte Ca$^{2+}$ levels were unaffected by interneuron single APs (Ca$^{2+}$ transient probability: from 0.09 ± 0.03 to 0.10 ± 0.03 after interneuron stimulation; p=0.92; paired *t* test; cf. [*Rózsa et al., 2015*]), they were increased by bursts of interneuron APs (Ca$^{2+}$ transient probability: from 0.08 ± 0.03 to 0.20 ± 0.03 after interneuron stimulation; p=0.007; paired *t* test) (Ca$^{2+}$ transient probability index: 0.49 ± 0.05; 58 astrocytes from nine slices; p<0.001; Wilcoxon rank-sum test) (*Figure 3B,C*), a phenomenon that was abolished by the GABA$_B$ receptor antagonist CGP55845 (Ca$^{2+}$ transient probability index: from 0.47 ± 0.10 to 0.05 ± 0.14 after CGP55845; 91 astrocytes from 10 slices; p=0.011; Wilcoxon rank-sum test; 0.034 ± 0.09 to 0.05 ± 0.14 after IN-stimulation in presence of CGP55845; p=0.824; Wilcoxon rank-sum test; *Figure 3C*). To test whether this interneuron-evoked astrocyte Ca$^{2+}$ signal was necessary for the synaptic potentiation, we pharmacologically prevented astrocyte Ca$^{2+}$ elevations while monitoring CA3-CA1 synaptic transmission (*Figure 3C–F*). After assessing that bursts of interneuron APs transiently increased the success rate of the recorded synapses, perfusion with thapsigargin (which depletes the internal Ca$^{2+}$ stores; 1 µM) abolished both the interneuron-induced astrocyte Ca$^{2+}$ signal (41 astrocytes from five slices; p=0.37; Wilcoxon rank-sum test; *Figure 3C*) and the synaptic potentiation (n = 9; p=0.46; paired *t* test; *Figure 3F*). To unambiguously down-regulate Ca$^{2+}$ signals selectively in astrocytes, we injected through the astrocytic recording pipette BAPTA (a Ca$^{2+}$ chelator; 40 mM) into the gap junction-coupled astrocytic network (*Di Castro et al., 2011*; *Navarrete and Araque, 2010*; *Serrano et al., 2006*) (*Figure 3C, D*). After BAPTA-loading into astrocytic network, bursts of interneuron APs failed to elevate astrocyte Ca$^{2+}$ (41 astrocytes from nine slices; p=0.26; Wilcoxon rank-sum test; *Figure 3C*) and to potentiate neurotransmission (n = 11; p=0.50; paired *t* test; *Figure 3D,F*). To discard indirect effects of BAPTA outside the astrocytic network on the IN-mediated modulation of synaptic transmission, a BAPTA-containing pipette was placed in *stratum radiatum* nearby to the stimulation electrode in absence of astrocyte recording. During 30 min of recording no changes in synaptic parameters were found (*Figure 3—figure supplement 1*), suggesting that the observed effect was not caused by the leakage of BAPTA in extracellular space and buffering Ca$^{2+}$, thus reducing transmitter release (cf. [*Serrano et al., 2006*]).

Since GABA$_B$ receptors are G protein-coupled receptors (*Meier et al., 2008*), and astrocytes GABA$_B$ receptors involve both Gi/o protein and inositol 1,4,5-trisphosphate (IP3) signalling pathways (cf.[*Mariotti et al., 2016*]), we used *Ip3r2$^{-/-}$* mice, in which G protein-mediated Ca$^{2+}$ mobilization in astrocytes is impaired (*Di Castro et al., 2011*; *Li et al., 2005*; *Navarrete et al., 2012*; *Petravicz et al., 2008*) to undoubtedly confirm the astrocytic involvement in interneuron-induced

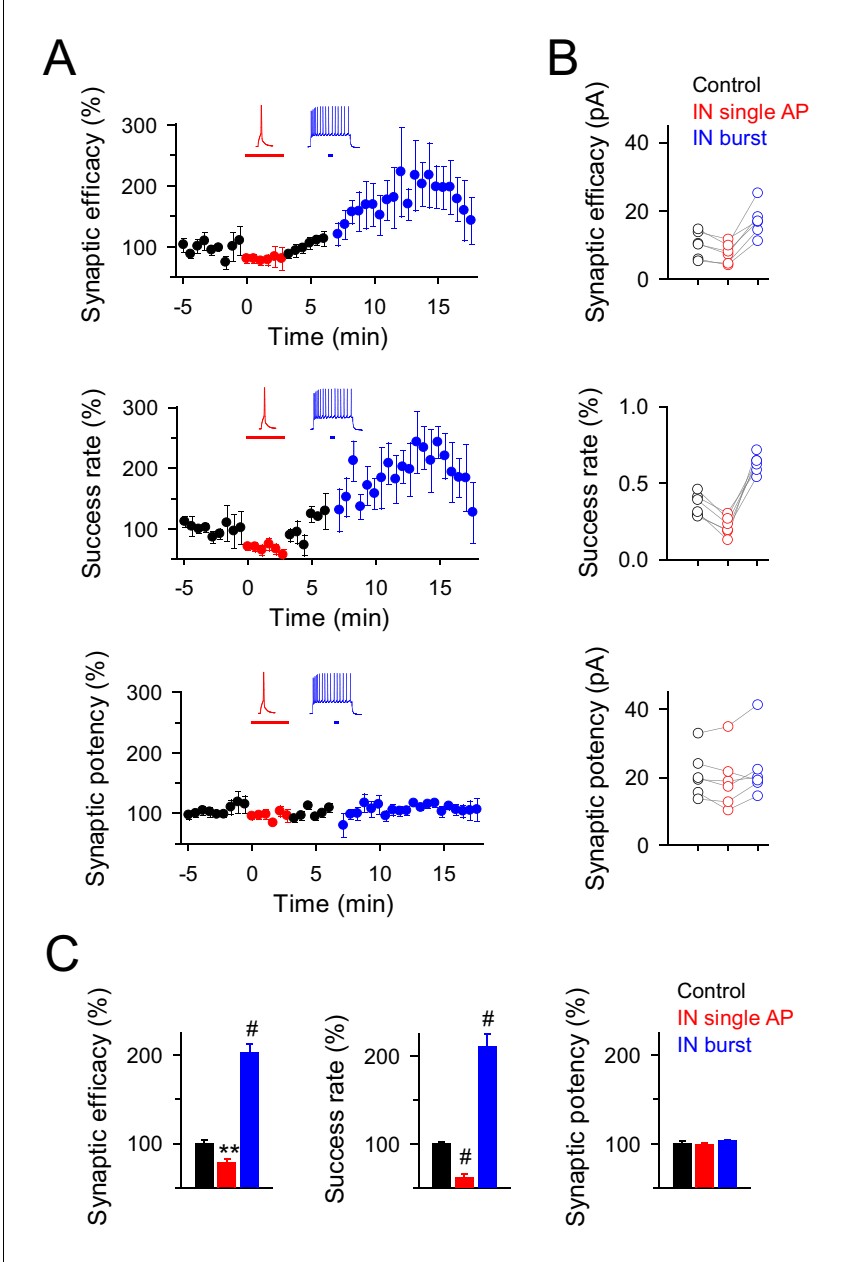

**Figure 2.** Interneuron activity regulates transmitter release at the same synapses depending on the firing rate. (**A**) Synaptic efficacy, success rate and synaptic potency before and after pairing SC stimuli with single (red) or bursts (blue) of interneuron APs ($n = 6$). Horizontal bars indicate the time of pairing. (**B**) Raw values of synaptic parameters plotted for both IN-stimulation conditions: control (black), IN single AP (red), and IN burst (blue) stimulation ($n = 6$). Note that the IN-mediated effects were independent on the initial values of synaptic parameters. (**C**) Relative changes of synaptic parameters from basal (black) elicited by pairing SC stimuli with single (red) or bursts of interneuron APs (blue) ($n = 6$). Synaptic efficacy: $p=0.008$ (red), $p<0.001$ (blue). Synaptic potency: $p=0.355$ (red), $p=0.407$ (blue). Success rate: $p<0.001$ (red), $p<0.001$ (blue); paired $t$ test. **$p<0.01$, #$p<0.001$; paired $t$ test.

synaptic potentiation. In slices from these animals, both astrocyte $Ca^{2+}$ signals (112 astrocytes from 14 slices; $p=0.43$; Wilcoxon rank-sum test; *Figure 3C*) and synaptic transmission parameters were unchanged by bursts of interneuron APs (8 out of 8 recorded pairs; $p=0.21$; paired $t$ test; *Figure 3E, F*), suggesting the contribution of astrocytic $GABA_B$ signaling to the interneuron-induced synaptic

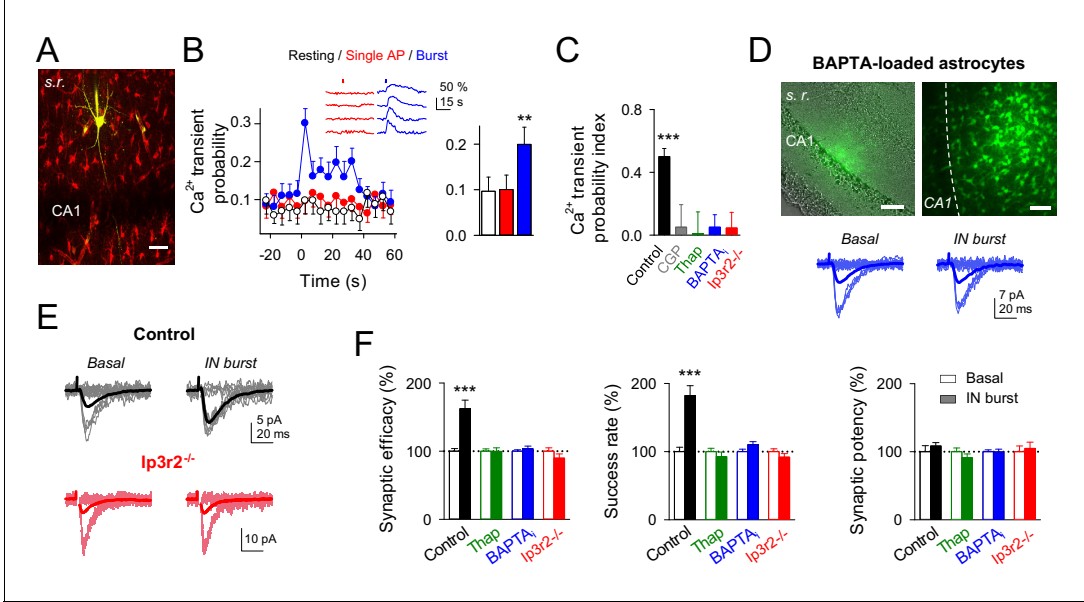

**Figure 3.** Interneuron-induced synaptic potentiation requires astrocyte Ca$^{2+}$ elevations. (**A**) Image of recorded interneuron (yellow) and SR101-labelled astrocytes (red) in CA1 region. Scale bar, 40 µm. (**B**) *Left,* astrocyte Ca$^{2+}$ transient probability over time (zero time corresponds to interneuron depolarization) in resting conditions (white), and after evoking single (red) or bursts (blue) of interneuron APs, and somatic Ca$^{2+}$ signals from four representative astrocytes in the field of view to single APs (red) and burst of interneuron APs (blue). Red and blue bars denote interneuron stimulation. *Right,* average values for those conditions. Single APs (red; p=0.92; paired *t* test), burst of interneuron APs (blue; p=0.007; 58 astrocytes, nine slices). (**C**) Ca$^{2+}$ transient probability index after bursts of interneuron APs in control (58 astrocytes, nine slices; p<0.001; Wilcoxon rank-sum test), CGP55845 (91 astrocytes, 10 slices; p=0.824; Wilcoxon rank-sum test), thapsigargin (41 astrocytes, five slices; p=0.37; Wilcoxon rank-sum test), BAPTA-loaded astrocytes (41 astrocytes, nine slices; p=0.2622; Wilcoxon rank-sum test), and *Ip3r2$^{-/-}$* mice (112 astrocytes, 14 slices; p=0.4319; Wilcoxon rank-sum test). (**D**) *Left,* merge DIC and fluorescence image of the CA1 pyramidal layer and stratum radiatum (s.r.) showing the location of the biocytin-filled astrocyte network (green). *Right,* maximal projection confocal image of the astrocytic syncytium revealed by biocytin-loading via whole-cell astrocyte recording, showing the distribution of biocytin-coupled astrocytes. Scale bars, 50 µm. *Bottom,* synaptic responses evoked by minimal stimulation (15 consecutive stimuli; light traces), and averaged EPSCs (50 consecutive stimuli; dark traces) before and after interneuron AP bursts in BAPTA-loaded astrocytes (blue). (**E**) Synaptic responses evoked by minimal stimulation (15 consecutive stimuli; light traces), and averaged EPSCs (50 consecutive stimuli; dark traces) before and after interneuron AP bursts in control (black) and *Ip3r2$^{-/-}$* mice (red). (**F**) Relative changes of synaptic parameters induced by bursts of interneuron APs (filled bars), in control (*n* = 6; p<0.001; paired *t* test), thapsigargin (*n* = 9), BAPTA-loaded astrocytes (*n* = 11), and *Ip3r2$^{-/-}$* mice (*n* = 8). See also *Figure 3—figure supplement 1*. **p<0.01, ***p<0.001; paired *t* test.

The following figure supplement is available for figure 3:

**Figure supplement 1.** BAPTA-containing pipette in the extracellular space does not affect synaptic responses.

potentiation. In contrast, the synaptic inhibition elicited by interneuron single APs was still present (success rate: 77.71 ± 6.16% from basal; *n* = 8; p=0.009; paired *t* test), indicating that GABA$_A$-mediated signaling was unaffected in these mice.

## Interneuron-induced synaptic potentiation requires presynaptic mGluR activation

Because the astrocyte Ca$^{2+}$ signal can stimulate the release of different gliotransmitters that modulate neuronal activity (*Perea and Araque, 2010*), we investigated the molecular mechanisms underlying the interneuron-induced astrocyte-mediated synaptic potentiation (*Figure 4*). Cholinergic signaling can increase synaptic transmission through activation of astrocytic muscarinic receptors (*Araque et al., 2002*; *Navarrete et al., 2012*; *Perea and Araque, 2005*; *Takata et al., 2011*). Antagonizing these receptors with atropine (50 µM) did not affect the interneuron-induced astrocyte Ca$^{2+}$ signal (n = 36 astrocytes from five slices, p=0.22; Wilcoxon rank-sum test), as well as the synaptic potentiation (n = 6, p=0.26; paired *t* test; *Figure 4B,C*). Endocannabinoid signaling might also enhance excitatory transmission directly by activation of astrocytic type one cannabinoid receptors

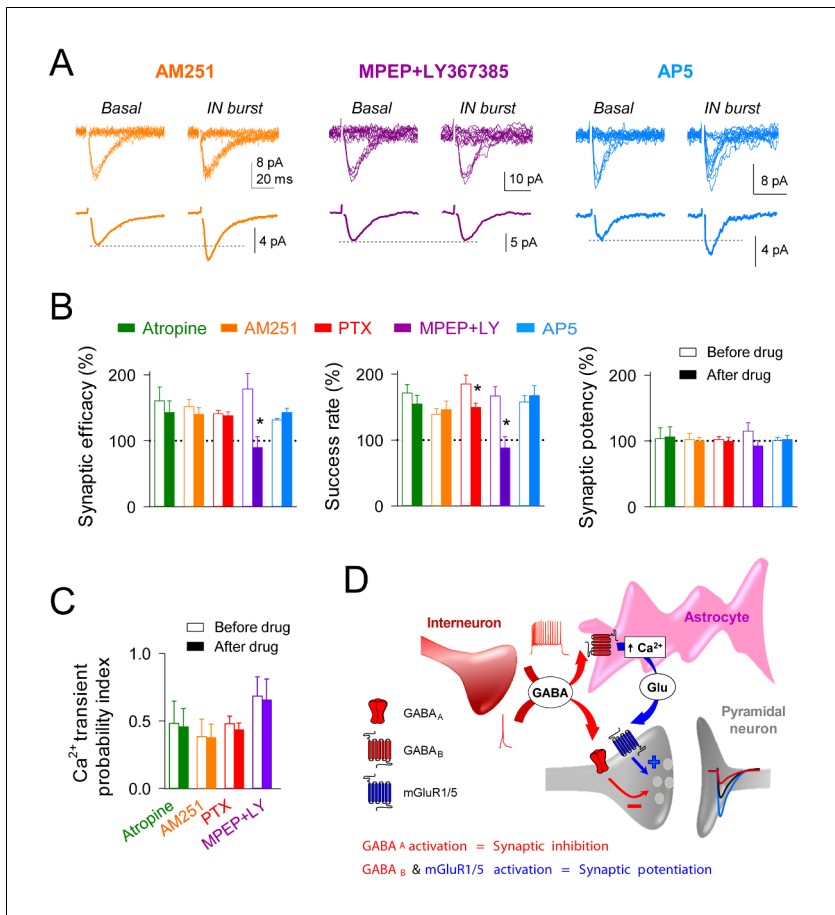

**Figure 4.** Interneuron-induced synaptic potentiation requires astrocytic GABA_B receptor and presynaptic mGluR activation. (**A**) Synaptic responses evoked by minimal stimulation (15 consecutive stimuli; top) and averaged EPSCs (bottom) before and after evoking bursts of interneuron APs in presence of AM251 (orange), MPEP+LY367385 (magenta), and AP5 (Blue). (**B**) Relative changes of synaptic parameters induced by bursts of interneuron APs before (open bars) and after (filled bars) receptor antagonist application ($n \geq 5$ neurons for each bar). *$p<0.05$; paired $t$ test. (**C**) Astrocyte $Ca^{2+}$ transient probability index after evoking bursts of interneuron APs, before (open bars) and after (filled bars) receptor antagonist application ($n \geq 5$ slices for each bar). (**D**) Scheme of the proposed mechanisms for GABA_A mediated synaptic inhibition and GABA_B-mGluR group I induced synaptic potentiation, respectively.

(CB1Rs) (*Navarrete and Araque, 2008*, *2010*) or indirectly by depressing GABA release from synaptic terminals (*Carlson et al., 2002*). The CB1R antagonist AM251 (2 μM) did not affect the interneuron-induced astrocyte $Ca^{2+}$ elevations (n = 32 astrocytes from five slices, p=0.13; Wilcoxon rank-sum test) or the synaptic potentiation (n = 5, p=0.17; paired $t$ test) (*Figure 4A–C*). Therefore, neither cholinergic nor endocannabinoid signaling were involved in the synaptic potentiation. Glutamate released by astrocytes regulates neurotransmission through activation of glutamatergic receptors (*Henneberger et al., 2010*; *Jourdain et al., 2007*; *Perea and Araque, 2007*), such as group I metabotropic glutamate receptors (mGluRs) (*Navarrete and Araque, 2010*; *Navarrete et al., 2012*; *Perea et al., 2014*; *Sasaki et al., 2012*). After assessing that bursts of interneuron APs increased the success rate of recorded synapses, perfusion with group I mGluR antagonists MPEP (50 μM) and LY-367385 (100 μM) blocked the synaptic potentiation (n = 6; p=0.02; paired $t$ test) (*Figure 4A,B*), without affecting the astrocyte $Ca^{2+}$ signal (39 astrocytes from five slices; p=0.76; Wilcoxon rank-sum test) (*Figure 4C*). Additionally, recent studies have reported the contribution of D-serine released by astrocytes to synaptic plasticity through NMDA receptor activation (*Henneberger et al., 2010*; *Takata et al., 2011*). However, the synaptic potentiation induced by interneuron activity was

unaffected by the perfusion of the NMDA receptor antagonist AP5 (50 µM; $n$ = 6; p=0.11; paired $t$ test) (*Figure 4A,B*); suggesting that EPSC modulation was independent of the astrocytic D-serine actions. Taken together, these results indicate that high interneuron activity enhance excitatory transmission through activation of astrocytic GABA$_B$ receptors, which stimulate Ca$^{2+}$-dependent glutamate release from astrocytes that activates group I mGluRs at excitatory terminals (*Figure 4D*).

## Interaction between interneuron and astrocyte signaling dynamically controls excitatory synaptic transmission

We next studied the dynamic interplay between interneuron, astrocyte activity and excitatory transmission to determine whether interneuron-mediated inhibition and potentiation could occur under particular conditions mimicking physiological hippocampal activity. Interneuron activity plays key roles in the appearance of certain oscillatory network activities, such as theta (3–12 Hz) and gamma frequency bands (25–100 Hz), which have relevant functions in coding neural information (*Lisman and Jensen, 2013*). Therefore, we stimulated the interneurons to elicit firing patterns that simulate those involved in these rhythms. Bursts of interneuron APs were elicited by depolarizing pulses (166 ms delivered at 3 Hz for 30 s) while continuously stimulating SC at 6 Hz. Thus, SC-evoked EPSCs were phase-locked at the interneuron depolarization (up-EPSC) and resting levels (down-EPSC) (*Figure 5A*, *inset*). The analysis of the responses showed that interneuron stimulation bursts evoked: (1) an overall potentiation of the synaptic efficacy mediated by an enhancement of the success rate at both up-EPSCs and down-EPSCs (success rate: 132.89 ± 5.32% and 150.23 ± 6.22% of basal, respectively; $n$ = 13; p<0.001 *vs.* basal; paired $t$ test) that lasted throughout the stimulus period, and (2) fast dynamic changes of the synaptic efficacy and success rate within each up/down cycle (i.e., depolarized or resting state of the interneuron) (*Figure 5A*), that is, up-EPSC values were significantly lower than those of immediately succeeding down-EPSCs (mean relative success rate difference: 17.34%; p=0.04; paired $t$ test) (*Figure 5B*). No significant changes in the synaptic potency (p=0.29; paired $t$ test; *Figure 5B*) were observed. Consistent with the mechanisms described above, the overall IN-evoked potentiation was unaffected by picrotoxin (down-EPSC, $n$ = 11; p<0.001; paired $t$ test), but blocked by CGP55485 (down-EPSC, $n$ = 11; p=0.97; paired $t$ test) or MPEP+LY367385 (down-EPSC, $n$ = 8; p=0.53; paired $t$ test) (*Figure 5A,B*), and absent in $Ip3r2^{-/-}$ mice (down-EPSC, $n$ = 7; p=0.06; paired $t$ test) (*Figure 5—figure supplement 1*), indicating that it requires GABA$_B$ receptor activation, astrocyte Ca$^{2+}$ signaling, and mGluR activation.

In contrast, differences in synaptic efficacy and the success rate in up-EPSC *vs.* down-EPSC were abolished by picrotoxin (mean relative success rate difference: 0.40%; $n$ = 11; p=0.79; paired $t$ test) (*Figure 5A,B*) but were still present in the rest of conditions. Indeed, mean relative differences for the success rate were 27.72% with CGP55845 ($n$ = 11; p=0.013; paired $t$ test) and 19.30% with MPEP+LY367385 ($n$ = 8; p=0.04; paired $t$ test) (*Figure 5B*), and 13.33% in $Ip3r2^{-/-}$ mice ($n$ = 7; p=0.01; paired $t$ test; *Figure 4B*), indicating that the reduced synaptic transmission during up-EPSCs relative to down-EPSCs was due to the GABA$_A$-mediated inhibition. Similar results were observed by identical interneuron activation when SC were stimulated at 3 Hz, independently phase-locked at either the up or down-state of interneuron activity (*Figure 5—figure supplement 2A,B*), showing the potentiation of synaptic transmission for both up- and down-EPSC and the significant differences in up-EPSC *vs.* down-EPSC, and indicating that the phenomena were unrelated to the SC stimulation frequency or the phase of interneuron stimulation. In contrast, no changes in synaptic transmission were observed in the absence of interneuron stimulation (*Figure 5—figure supplement 2C,D*), showing that synaptic stimulation per se did not account for the potentiation observed and that interneuron-astrocyte signaling was necessary to induce the enhancement of excitatory synaptic transmission. In summary, bursts of interneuron activity induced a dynamic modulation of CA3-CA1 synaptic transmission consisting on an overall steady potentiation superimposed with faster transitions within each cycle of up and down interneuron states. While the latter and faster effect is due to GABA$_A$-mediated inhibition of transmitter release, the former and sustained modulation is mediated by astrocytic GABA$_B$ receptors, which stimulate the Ca$^{2+}$-dependent release of glutamate that activate presynaptic mGluRs at the excitatory terminals.

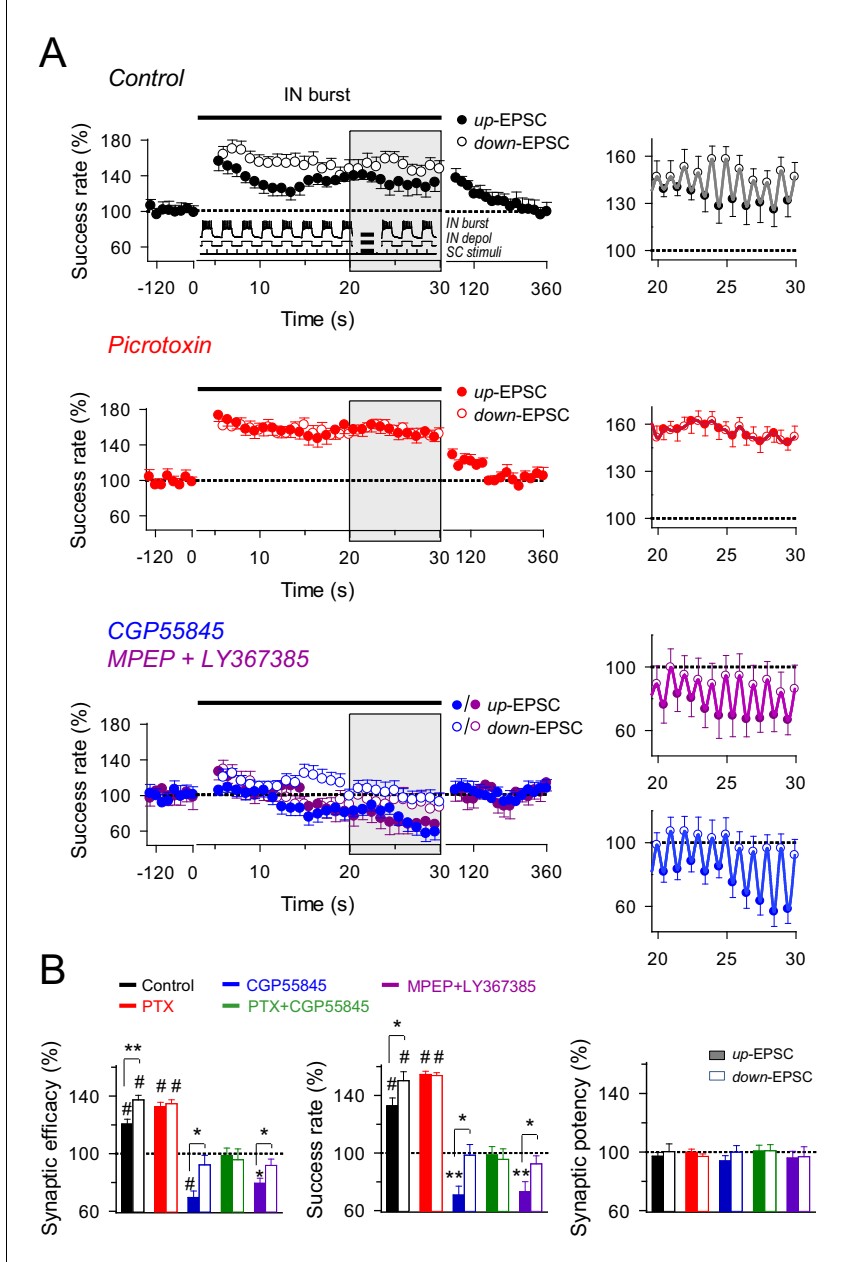

**Figure 5.** Effects of dynamic interplay between interneuron and astrocyte activity on excitatory synaptic transmission. (**A**) *Left*, averaged relative success rate of up-EPSCs and down-EPSCs over time evoked by a stimulation paradigm (*inset*) consisting of SC stimuli at 6 Hz and bursts of interneuron APs elicited by 90 depolarizing pulses (166 ms delivered at 3 Hz for 30 s), in control (black; n = 13), picrotoxin (PTX, red; n = 11), CGP55845 (blue; n = 11), and MPEP+LY (magenta; n = 8). Note that SC stimuli were phase-locked at the interneuron depolarization (up-EPSC) or resting level (down-EPSC). Each point represents the simple moving average of 15 consecutive EPSCs (note the corresponding initial gap at interneuron stimulation). Horizontal bars indicate the time of pairing. *Right*, expanded view of gray-shaded areas. (**B**) Relative changes of synaptic parameters relative to basal of up-EPSCs (closed bars) and down-EPSCs (open bars) in control and after receptor antagonist application (PTX + CGP55845; n = 9). See also *Figure 5—figure supplements 1* and *2*. *p<0.05, **P<0.01, #p<0.001; paired *t* test.

The following figure supplements are available for figure 5:

**Figure supplement 1.** GABA$_A$ and GABA$_B$ interplay in the astrocyte-interneuron modulation of the excitatory synaptic transmission.

*Figure 5 continued on next page*

*Figure 5 continued*

**Figure supplement 2.** Effects of phase-locked activity between interneuron and excitatory synaptic activity.

## Astrocyte GABA_B receptors are involved in hippocampal theta and gamma oscillations in vivo

Because GABA$_B$ receptors are present in neurons, both pre- and postsynaptically (*Vigot et al., 2006*), and in astrocytes (*Meier et al., 2008*), to further support the astrocytic function of GABA$_B$ receptors, we generated genetically modified mice with conditional ablation of GABA$_B$ receptors specifically in astrocytes (GB1-cKO mice). Deletion of the receptor subunit gene *Gabbr1* is sufficient to completely abolish GABA$_B$ receptor function, even when the GABBR2 subunit is expressed as well. We crossed GLAST-CreERT2 knockin mice (*Mori et al., 2006*) with *Gabbr1*$^{fl/fl}$ mice (*Haller et al., 2004*) (*Figure 6A*). Mice with astrocyte-specific GABA$_B$ receptor ablation were investigated 6 to 8 weeks after induction of gene recombination by intraperitoneal injection of tamoxifen. qRT-PCR of genomic DNA revealed 28.5% of *Gabbr1* gene recombination (*n* = 7 for both GB1-cKO and control mice; p=0.003; unpaired *t* test; *Figure 6B*), a number reflecting the percentage of astrocytes in the hippocampus. Due to the high abundance of neuronal expression, the mRNA levels were not significantly reduced in GB1-cKO mice (*n* = 3 for both GB1-cKO and control mice; p=0.24; unpaired *t* test; *Figure 6B*). Quantifying the co-localization of astroglial *GLAST* and *Gabbr1* immuno-labels (*Cordelières and Bolte, 2014*), however, demonstrated a significant reduction of *Gabbr1* at the protein level (9 sections from 2 GB1-cKO, 11 sections from two control mice; 2-sided; p<0.001; unpaired *t* test; *Figure 6B*), although the spatial resolution in single optical sections of a confocal laser-scanning microscope might underestimate the *Gabbr1* removal from the fine astrocyte processes.

The functional presence of GABA$_B$ receptors was monitored in astrocytes and neurons by local application of the selective GABA$_B$ agonist baclofen. Astrocytes from littermate control mice showed baclofen-induced Ca$^{2+}$ responses, as well as ATP-induced Ca$^{2+}$ responses, suggesting competent receptor-mediated intracellular Ca$^{2+}$ signaling pathways in these astrocytes (Ca$^{2+}$ transient probability index: 0.53 ± 0.10 and 0.51 ± 0.09, for baclofen and ATP, respectively; 35 astrocytes from seven slices, two mice; p<0.001; Wilcoxon rank-sum test) (*Figure 6D,E*). Additionally, astrocytes from littermate wild-type mice also showed the interneuron-induced Ca$^{2+}$ responses after interneuron stimulation (Ca$^{2+}$ transient probability index: 0.50 ± 0.07, 84 astrocytes from five slices, three mice; p<0.001; Wilcoxon rank-sum test; *Figure 6D,E*). In contrast, astrocytes from GB1-cKO mice (*Figure 6C*) showed a remarkable absence of Ca$^{2+}$ responsiveness to baclofen while ATP-evoked Ca$^{2+}$ signaling was unaffected (Ca$^{2+}$ transient probability index: 0.13 ± 0.08; p=0.12, with baclofen and 0.61 ± 0.03; p<0.001, with ATP; Wilcoxon rank-sum test; 82 astrocytes and 88 astrocytes from eight slices, respectively; three mice) (*Figure 6D,E*). According to these data, interneuron stimulation did not evoke either astrocyte Ca$^{2+}$ signaling (Ca$^{2+}$ transient probability index: 0.12 ± 0.08; 71 astrocytes from eight slices; p=0.14; Wilcoxon rank-sum test; *Figure 6E*) or the potentiation of excitatory synaptic transmission in GB1-cKO mice (success rate: 94.75 ± 19.75 of basal; *n* = 4; p=0.49; paired *t* test) (*Figure 6G,H*). In contrast, wild-type mice showed the synaptic potentiation induced by interneuron stimulation (success rate: 148.01 ± 17.25 of basal, *n* = 5; p<0.001; paired *t* test), which was abolished in the presence of the GABA$_B$ receptor antagonist CGP55845 (success rate: 103.94 ± 11.11 from basal; *n* = 3; p=0.75; paired *t* test) (*Figure 6G,H*).

We next evaluated the neuronal responses evoked by baclofen in CA1 pyramidal cells from wild-type and GB1-cKO mice in the presence of TTX (1 µM). Local application of baclofen (2 mM, 10 s) induced outward currents either in wild-type (13.73 ± 2.12 pA; n = 8 baclofen puffs; three neurons) and GB1-cKO neurons (10.76 ± 1.0 pA; n = 9 baclofen puffs; three neurons; p=0.398; Wilcoxon rank-sum test), which were both abolished by CGP55845 (*Figure 6F*). These results indicate that the selective deletion of the receptor subunit gene *Gabbr1* in astrocytes did not affect the neuronal GABA$_B$ receptor activity. Therefore, these data indicate that GABA$_B$-induced Ca$^{2+}$ elevations in astrocytes are required for the interneuron-evoked potentiation of excitatory synaptic transmission.

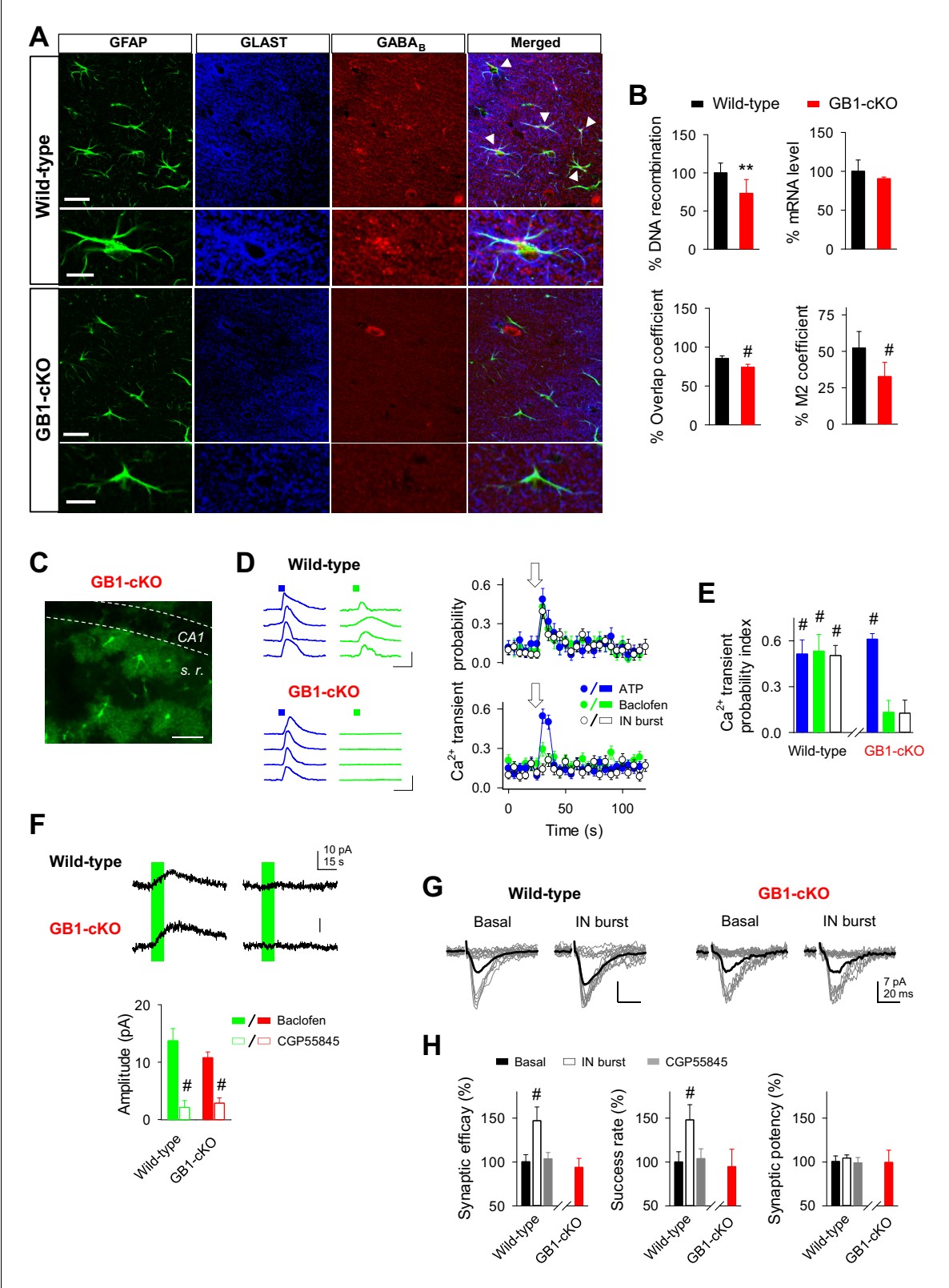

**Figure 6.** GABA$_B$ receptor knockout in astroglial cells (GB1-cKO mice) impairs interneuron-mediated synaptic potentiation. (**A**) Confocal laser-scanning micrographs of hippocampal sections from wild-type (top row) and GB1-cKO (GLAST-CreERT2xGABBR1$^{fl/fl}$; bottom row) wildtype mice immunostained for the glial fibrillary acidic protein GFAP, astroglial glutamate transporter GLAST, and the GABA$_B$ receptor GABBR1. Please note the white arrow heads in wild-type that point towards co-localization of the GABA$_B$ receptor on astroglial GFAP-positive structures. In contrast, such locations of co-

*Figure 6 continued on next page*

*Figure 6 continued*

localization are largely missing in the GB1-cKO mice. Scale bars, 30 µm; inset, 15 µm. (**B**) *Top left*, qRT-PCR of hippocampal genomic DNA reveals a reduction of the *Gabbr1*$^{fl/fl}$ alleles by 28.5% (p=0.003; unpaired *t* test), representing the percentage of astrocytes in the hippocampus. *Top right*, quantification of *Gabbr1* mRNA levels by qRT-PCR does not show a reduction of the *Gabbr1* message (p=0.25; unpaired *t* test), as expected by the high levels of neuronal versus glial expression. Quantification of the immunolabels for *Glast* and *Gabbr1* were determined by the ImageJ plugin JACoP v.2 by calculating the overlap (p<0.001; unpaired *t* test; *bottom left*) and Mander's M2 coefficients (p<0.001; unpaired *t* test; *bottom right*), both indicate a significant reduction of astroglial *Gabbr1* expression (9 sections from 2 GB1-cKO and 11 sections from two control mice). (**C**) Confocal image from CA1 region of GB1-cKO mice showing the endogenous expression of GCaMP3 in astrocytes lacking *Gabbr1* (in green). Scale bar, 60 µm. (**D**) Intracellular $Ca^{2+}$ signals induced by local agonist application of ATP (blue) and baclofen (green) from four representative astrocytes in wild-type and GB1-cKO mice, and astrocyte $Ca^{2+}$ transient probability over time induced by agonist stimulation or after evoking bursts of interneuron APs (white). Arrow, green and blue squares denote ATP, baclofen, or interneuron stimulation. Scale bar, 100%, 15 s. (**E**) $Ca^{2+}$ transient probability index after astrocyte stimulation in wild-type (n = 35 astrocytes from seven slices; p<0.001; Wilcoxon rank-sum test) and GB1-cKO mice. GB1-cKO astrocytes failed to increase intracellular $Ca^{2+}$ in response to baclofen (n = 88 astrocytes from eight slices; p=0.12; Wilcoxon rank-sum test), and IN stimulation (n = 71 astrocytes from eight slices; p=0.14; Wilcoxon rank-sum test), but were activated by ATP (n = 82 astrocytes from eight slices; p<0.001; Wilcoxon rank-sum test). (**F**) Baclofen-evoked currents in CA1 pyramidal neurons from wild-type and GB1-cKO mice before and after CGP55845 application (Wild-type: from 13.73 ± 2.12 to 2.09 ± 1.22 pA, before and after CGP55845; n = 7; paired *t* test; p<0.001; GB1-cKO: 10.76 ± 1.0 to 2.85 ± 0.89 pA; before and after CGP55845; n = 8; p<0.001). (**G**) Synaptic responses evoked by minimal stimulation (15 consecutive stimuli; gray traces), and averaged EPSCs (50 consecutive stimuli; black traces) before and after evoking bursts of interneuron APs in wild-type and GB1-cKO mice. (**H**) Relative changes of synaptic parameters from basal (black bars) induced by bursts of interneuron APs (white bars; n = 5; p<0.001; paired *t* test), and after CGP55845 application (n = 3; p=0.38; paired *t* test) in wild-type and GB1-cKO mice (n = 4; p=0.38; paired *t* test). **p<0.01; # p<0.001.

In order to evaluate the impact of astrocytic GABA$_B$ receptor expression deficiency to the coding properties of cortical networks, we performed in vivo recordings in anesthetized GB1-cKO mice and wild-type littermates. We examined the basic properties of local field potentials (LFP) in the dorsal hippocampus in resting conditions (*Figure 7—figure supplement 1*) and after whisker stimulation (*Figure 7A*). Analysis of the LFP responses and power spectrum revealed that sensory stimulation boosted the theta and gamma components in control littermates, and significant changes occurred in those oscillatory neuronal responses in mice with down-regulated expression of astrocytic GABA$_B$ receptors (*Figure 7B,C*). Both theta band (peak 4–8 Hz; p=0.007; unpaired *t* test) and low gamma band activities (peak 30–50 Hz; p=0.042; Wilcoxon rank-sum test) were partially reduced in GB1-cKO mice (*Figure 7C*), indicating a significant role of astrocytic GABA$_B$ receptors in these network oscillations, which are related to cognitive and behavioral tasks (*Buzsáki and Moser, 2013*). Theta–gamma coupling is thought to have an important function regulating hippocampal–cortical and sub-cortical communication during learning, episodic memory and recall tasks (*Hanslmayr et al., 2016*; *Tort et al., 2009*). We therefore analyzed the cross-frequency coupling between theta phase and gamma amplitude in resting and after whisker stimulation (*Figure 7D*). We found that the magnitude of phase-amplitude coupling, measured as phase-locking value (PLV), was enhanced in control litter-mates after sensory stimulation (PVL= from 0.30 ± 0.03 to 0.50 ± 0.07; n = 24 epochs; p=0.017; paired *t* test; *Figure 7E*) as reported previously in behaving rodents (*Csicsvari et al., 2003*; *Tort et al., 2009*). In contrast, the coupling between theta and gamma oscillations was disrupted in GB1-cKO mice, which did not show changes in PVL index after whisker stimulation (PVL= from 0.39 ± 0.03 to 0.42 ± 0.04; n = 36 epochs; p=0.62; paired *t* test; *Figure 7E*), indicating that ablation of GABA$_B$ receptors in astrocytes weaken either hippocampal theta and gamma rhythms as well as their coupling. Additionally, the impact of astrocyte-GABAergic signaling was revealed under stimu-lus driven-hippocampal activity conditions (i.e., whisker stimulation), but no differences were observed in resting conditions (*Figure 7—figure supplement 1*), suggesting a state-dependent astrocyte neuromodulation of hippocampal rhythms in vivo. In summary, astrocytic GABA$_B$ receptors are involved in an oscillatory brain activity in vivo, contributing to theta and low gamma waves in stimulus-driven conditions.

## Discussion

The operation of neuronal networks crucially depends of a fast time course of signaling in inhibitory interneurons. Our results show that GABAergic signaling dynamically impacts excitatory transmission in an activity- and time-dependent manner that is controlled by astrocytes. While excitatory

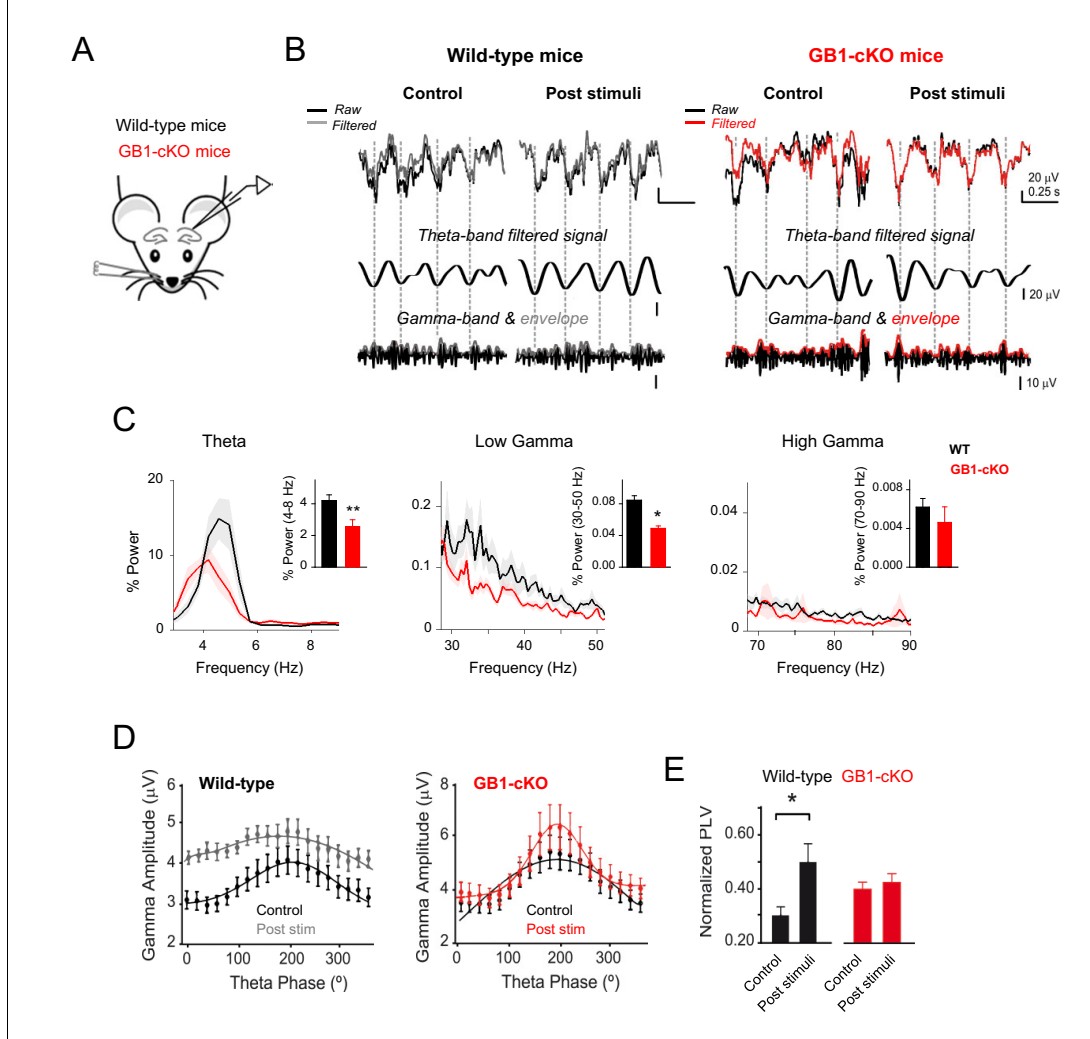

**Figure 7.** Astrocyte GABA$_B$ receptors participate in hippocampal theta and gamma oscillations in vivo. (**A**) Schematic illustration of the hippocampal recording configuration and whisker stimulation in anesthetized mice. (**B**) Representative LFP recordings and corresponding analysis of theta-phase and gamma-amplitude relation for control littermate (left) and GB1-cKO (right) mice. The raw signals (black) were high-pass filtered (1st row; grey and red, respectively) and then computed to extract the theta phase (second row) and gamma envelope (third row; grey and red, respectively) for control and GB1-cKO mice. (**C**) Normalized LFP power spectrum analysis for theta (4–8 Hz) and gamma frequencies (30–50 Hz, low gamma; 70–90 Hz, high gamma) in control littermate (*n* = 6) and GB1-cKO mice (*n* = 6) after whisker stimulation. *Inset,* Relative power changes for GB1-cKO and control littermate mice (theta band, p=0.007; unpaired *t* test; low gamma, p=0.042; Wilcoxon rank-sum test; and high gamma oscillations, p=0.738; Wilcoxon rank-sum test). (**D**) Gamma-amplitude modulation by theta-phase for wild-type (left) and GB1-cKO mice (right), before (control) and after whisker stimulation (post stim). Note the enhancement of theta-gamma coupling in wild-type after stimulus that does not appeared in GB1-cKO mice. (**E**) Normalized Phase-Lock Value (PLV), either in control and after stimulation for wild-type (p=0.017; paired *t* test), and GB1-cKO mice (p=0.62; paired *t* test). See also ***Figure 7— figure supplement 1***. *p<.05; **p<0.01. .

The following figure supplement is available for figure 7:

**Figure supplement 1.** Resting activity in GB1-cKO and control littermate mice.

neurotransmission was inhibited by interneuron single APs through activation of GABA$_A$ receptors (characterized by fast kinetics) (***Farrant and Nusser, 2005***), it was enhanced by bursts of interneuron APs through additional and concurrent slower mechanisms (i.e., astrocyte GABA$_B$ receptor activation, G-protein-mediated intracellular Ca$^{2+}$ mobilization, astrocytic glutamate release, and presynaptic group I mGluR activation) that persisted after the interneuron burst (i.e., during the interneuron down-state). Therefore, different patterns of interneuron activity determine diverse consequences on

excitatory synaptic transmission by direct interneuron-principal cell mechanisms (fast inhibition) and by novel interneuron-astrocyte signaling mechanisms (slow potentiation). In the latter mechanisms, astrocytes decode interneuron activity and transform inhibitory GABAergic signals into excitatory glutamatergic signals that enhance synaptic transmission. In addition, we found a subset of synapses that express both phenomena, switching from inhibition to slow potentiation according to the inhibitory GABAergic tone. Thus, the existence of new mechanisms to fine-tune the output of local synapses by astrocyte activity might contribute to control the excitation-inhibition balance at the hippocampal circuits.

GABA$_B$ receptors have been proposed to inhibit transmitter release and modulate plasticity via presynaptic and postsynaptic mechanisms (*Ulrich and Bettler, 2007*). Thus, the activation of GABA$_B$ autoreceptors requires strong stimulus intensities, consistent with their distant location from the release sites and probably requires pooling of synaptically released GABA to be activated (*Ulrich and Bettler, 2007*). Here, we show that the activation of interneruons evoked synaptic inhibition independent of GABA$_B$ signaling, because it was present after their blockage with CGP55845 but was abolished by the GABA$_A$ receptor antagonist picrotoxin (*Figures 1*, *4* and *5*). Although, GABA spillover and activation of presynaptic GABA$_B$ cannot be totally excluded, the pharmacological data and results obtained from *Ip3r2$^{-/-}$* mice, where GABA$_B$ receptors in neurons and astrocytes are intact, and the new transgenic mouse lacking GABA$_B$Rs specifically in astrocytes (*Figure 6*) indicate that GABA$_B$ autoreceptors do not contribute to the observed inhibition of synaptic transmission.

The activity of hippocampal interneurons has been shown to activate astrocytes that induce a long-lasting enhancement of inhibitory synaptic transmission through glutamate release and activation of kainate receptors in inhibitory terminals (*Kang et al., 1998*). Our results indicate that the glutamate released from astrocytes stimulated by interneurons can also access mGluRs in excitatory synapses to transiently enhance their synaptic efficacy, suggesting that a single gliotransmitter may have multiple effects depending on the site of action. In addition to glutamate, hippocampal astrocytes may also release ATP, which is converted to adenosine that depress synaptic transmission through activation of A1 adenosine receptors (*Andersson et al., 2007*; *Chen et al., 2013*; *Pascual et al., 2005*; *Serrano et al., 2006*). This mechanism has been proposed to be triggered by high frequency stimulation of SC that sequentially activates interneurons and astrocytes leading to the heterosynaptic depression in the hippocampal CA1 region (*Andersson et al., 2007*; *Serrano et al., 2006*). Then, astrocytes immersed in the same circuit may be stimulated by interneurons to release different gliotransmitters (i.e., glutamate or ATP) that influence synaptic transmission in different forms. The specific circumstances leading to the release of particular gliotransmitters are unknown. While we directly stimulated interneurons to elicit specifically identified firing patterns, the activity evoked in interneurons during high frequency stimulation of SC was unknown. Since our results indicate that astrocytes decode interneuron activity, it is possible that the different regulatory effects were due to differences in the astrocyte signaling evoked by interneurons. This represents an additional example of the importance of the context-specificity of signaling in the reciprocal communication between neurons and astrocytes (for a detailed discussion, see [*Araque et al., 2014*]). Because the molecular signaling governing interneuron-astrocyte mediated effects shown here were studied in juvenile animals, and considering the developmental receptor expression profiles (*Sun et al., 2013*), whether these molecular pathways and complex features of interneuron-astrocyte signaling are conserved in the adult brain need further attention. In addition, present data cannot discard that residual Ca$^{2+}$ events might occur in the fine process of the GB1-cKO astrocytes, as they have been found for *Ip3r2$^{-/-}$* mice (*Srinivasan et al., 2015*); however, the existence of those events would not be sufficient to induce the synaptic potentiation observed after interneuron stimulation (*Figure 3E,F*, and *Figure 6G,H*). Thus, these data suggest that although the astrocyte-synaptic interactions might primarily take place at the astrocyte processes, the synaptic plasticity induced by interneuron-astrocyte communication is a highly regulated phenomenon that requires the active contribution of astrocyte somatic Ca$^{2+}$ signaling.

Numerous functions of interneurons crucially depend on the fast and temporally precise conversion of an excitatory synaptic input into an inhibitory synaptic output. As a result, interneurons provide 'phasic' inhibition to the neuronal network, which is involved in the emergence of fast brain rhythms (*Csicsvari et al., 2003*) and synaptic plasticity during critical periods of circuit formation (*Hensch, 2005*) that jointly contribute to the maturation of cognitive functions. Here, we imposed to

interneurons an oscillatory activity of 3 Hz with intra-faster oscillations of 35–50 Hz (*Figure 5A*). That rhythm, which is similar to physiological slow delta/theta (2–8 Hz) and low frequency gamma (25–40 Hz) bands found in the hippocampus, evoked fast episodes of inhibition nested within a long lasting enhancement of excitatory transmission. To unambiguously determine the locus and contribution of GABA$_B$ receptors to the synaptic transmission and network coding properties, and considering that GABA$_B$ receptors are ubiquitous, we took advantage of a conditional astrocyte-specific GABA$_B$ knockout mouse. The data shown here indicate that GB1-cKO mice showed a down-regulated GABA$_B$-astrocyte Ca$^{2+}$ signaling and the absence of the interneuron-mediated potentiation of excitatory synaptic transmission. In vivo recordings in anesthetized mice remarkably displayed changes in the hippocampal oscillatory activity pattern. We found that theta and gamma oscillations that are associated with cognitive processes, as well as the coupling between these rhythms were compromised in mice lacking GABA$_B$-astrocyte signaling, indicating a critical role of astrocyte signaling in higher-order information coding. Thus, recent studies have shown how astrocyte activity can impact the state of neuronal circuits by regulating the generation of neuronal UP states (*Poskanzer and Yuste, 2011*), and it has been related to brain rhythms (*Poskanzer and Yuste, 2016*), such as slow cortical oscillations (<1 Hz) associated with nonrapid eye movement (NREM) sleep (*Fellin et al., 2009*). Disruption of astrocytic vesicular release has been found crucial for gamma oscillatory hippocampal activity with significant impact in recognition memory tasks (*Lee et al., 2014*). Therefore, giving the important role of interneuron activity and hippocampal oscillations in coding neural information, such as the control of the theta phase-modulation of gamma power that correlates with memory performance (*Buzsáki and Moser, 2013*; *Lisman and Jensen, 2013*), the signaling between interneurons and astrocytes, which provides novel properties to the interneuron effects on excitatory synapses within the network, adds more complexity to neuronal information processing.

Additionally, the GABA$_B$ receptor pharmacological blockade has been shown to enhance cognitive task performance by activating hippocampal theta and gamma rhythms in behaving rats (*Leung and Shen, 2007*); in contrast, a recent study shows that GABA$_B$ deletion in glutamatergic terminals disrupts the acquisition and learning of hippocampal tasks, demonstrating their contribution to learning-dependent synaptic changes and network dynamics (*Jurado-Parras et al., 2016*). Here, present data from astrocyte *Gabbr1* knockout mice show a partial but significant decrease of stimulus-induced theta-gamma oscillations and coupling, and highlight the intricate roles of GABA$_B$ receptors in regulating the neural network operation considering their specific cellular targets. Together, these data suggest that astrocytes might be directly related to critical brain rhythms and their cognitive functions (*Lee et al., 2014*). Considering that the promoter used GLAST can be expressed by progenitor cells in adult brain (*Slezak et al., 2007*), we cannot rule out a partial contribution of derived cells from those progenitors to the in vivo observed responses. Some evidence have shown that acute brain slices might undergo hypoxic conditions causing reactive changes in astrocytes (*Takano et al., 2014*), and a downregulation of GABA$_A$ receptor expression (*Zonouzi et al., 2015*); however, since the study of molecular individualities of the interneuron-astrocyte signaling show limitations that need to be explored ex vivo, these associated alterations and their potential influence cannot not be excluded from the observed responses.

The current view for the mechanisms underlying brain diseases is largely based on neuronal dysfunctions, but increasing evidence suggests that also disturbances of astrocyte-neuron interactions are related to brain disorders (*Seifert et al., 2006*; *Takano et al., 2009*). Because alterations of the excitatory/inhibitory balance might underlie different brain states and diseases, such as epileptic activity, schizophrenia, and mood disorders, present results showing astrocytic contribution to the excitatory drive, i.e., transformation of inhibitory signals into an excitatory enhancement, indicate that interneuron-astrocyte signaling might be involved in the excitatory/inhibitory unbalance present in particular brain states.

Taking together, present findings reveal novel and unexpected consequences of interneuron signaling in neuronal network activity through stimulation of astrocytes. Astrocytes decode the temporal activity of neurons and transform neuronal signals to impact circuit function through novel mechanisms based on different signaling and time scales. Thus, interneuron activation of astrocytes through the control of oscillatory activity is directly involved in the coding of neuronal circuits and their functional properties, suggesting that brain the function results from the dynamic interplay of Astrocyte-Neuron networks.

## Materials and methods

All the procedures for handling and sacrificing animals followed the European Commission guidelines for the welfare of experimental animals (2010/63/EU), US National Institutes of Health and the Institutional Animal Care and Use Committee at the University of Minnesota (USA). The use of astrocyte-specific GABBR1 knockout mice was approved by the Saarland state´s 'Landesamt für Gesundheit und Verbraucherschutz' in Saarbrücken/Germany (animal license number 72/2010). Animals of both genders were used, and were housed in standard laboratory cages with ad libitum access to food and water, under a 12:12 hr dark-light cycle in temperature-controlled rooms.

### Hippocampal slice preparation

Hippocampal slices were obtained from Wistar rats (P13–18), C57BL/6 wild-type mice, $Ip3r2^{-/-}$ (MGI:3640970) mice (P13–18), astrocyte-specific GABAB receptor knockout mice (GB1-cKO) and control littermates (P45–55), and PV-Cre (JAX #008069) mice backcrossed to a Cre-responsive reporter line (Ai9-rcl-tdTomato transgenic mice; JAX #007909). $Ip3r2^{-/-}$ mice were generously donated by Dr J Chen (University of California San Diego, CA, USA) (Li et al., 2005). Because no significant differences were found in the reported effects, results obtained in slices from rats and C57BL/6 wild-type mice were pooled together. Mice were anesthetized and decapitated. The brain was rapidly removed and placed in ice-cold artificial cerebrospinal fluid (ACSF). Slices (350–400 μm) were incubated during >1 hr at room temperature (22–24°C) in ACSF containing (in mM): NaCl 124, KCl 2.69, $KH_2PO_4$ 1.25, $MgSO_4$ 2, $NaHCO_3$ 26, $CaCl_2$ 2, and glucose 10, and was gassed with 95% $O_2$/5% $CO_2$ (pH = 7.3). Slices were then transferred to an immersion recording chamber and superfused with gassed ACSF. Cells were visualized under an Olympus BX50WI microscope (Olympus Optical, Tokyo, Japan).

### Electrophysiology

Electrophysiological recordings from interneurons, CA1 pyramidal neurons and astrocytes were made using the whole-cell configuration of the patch-clamp technique. Patch electrodes had resistances of 3–10 MΩ when filled with the internal solution that contained (in mM): K-Gluconate 135, KCl 10, HEPES 10, $MgCl_2$ 1, ATP-$Na_2$ 2 (pH = 7.3, with KOH). Recordings were obtained with PC-ONE amplifiers (Dagan Corporation, Minneapolis, MN, USA). Fast and slow whole-cell capacitances were neutralized and series resistance was compensated (≈70%). Recordings were rejected when the access resistance increased >20% during the experiment. Recordings from CA1 pyramidal neurons were performed in voltage-clamp conditions and the membrane potential was held at −70 mV to record Schaffer collateral (SC) afferents-evoked EPSCs. CA1 interneurons were recorded in current-clamp conditions. Signals were fed to a Pentium-based PC through a DigiData1440 interface board (Molecular Devices, Sunnyvale, CA, USA). The pCLAMP 10 software (Molecular Devices) was used for stimulus generation, data display, acquisition, storage and analysis. Experiments were performed at room temperature (22–24°C). For astrocyte network loading, the holding potential was −80 mV. BAPTA (40 mM) and biocytin (0.1%) intracellular astrocyte filling was performed for 20–30 min (internal solution contained (in mM): BAPTA-$K_4$ 40, NaCl, 8, $MgCl_2$ 1, HEPES 10, GTP-tris salt 0.4, ATP-$Na_2$ 2; pH = 7.3, with KOH.

Slices were then fixed and biocytin was revealed by Alexa488-Streptavidin (Figure 3D), showing the wide area covered by the intracellular biocytin loading, and confirming the broad downregulation of $Ca^{2+}$ signals by BAPTA intracellular filling astrocytes (cf.[Poskanzer and Yuste, 2011; Serrano et al., 2006]).

Baclofen (2 mM) was locally applied through a micropipette (10 s duration) in the presence of TTX (1 μM) to induce GABA$_B$-mediated currents in CA1 pyramidal neurons (holding potential set to −30 mV) from astrocyte-specific GABA$_B$ receptor knockout mice (GB1-cKO) and control littermate mice.

Minimal stimulation was achieved using theta capillaries (2–5 μm tip diameter) filled with ACSF, and placed in the stratum radiatum to stimulate SC afferents. Single pulses (250 μs duration) or paired pulses (50 ms interval) were delivered at 0.5 Hz by stimulator S-900 (Dagan). The stimulus intensity (1–15 mA) was adjusted to meet the conditions that putatively stimulate a single or very few synapses (Navarrete and Araque, 2010; Navarrete et al., 2012; Perea and Araque, 2007), and was unchanged for the entire experiment. The recordings that did not meet these criteria and

synapses that did not show amplitude stability of EPSCs were rejected. The synaptic parameters analyzed were: synaptic efficacy (mean EPSC peak amplitude of all evoked responses, including failures), synaptic potency (mean EPSC peak amplitude of successful responses when failures are excluded), the success rate of neurotransmitter release (calculated as the ratio between the number of effective EPSCs divided by the total number of stimuli), and paired-pulse ratio (PPR, 50 ms pulse interval) (*Fernández de Sevilla et al., 2002*; *Stevens and Wang, 1994*). The responses and failures were identified by visual inspection and PPF was quantified as second EPSC/1 st EPSC. Basal values were recorded 10 min before the stimulus (e.g., *Figure 1—figure supplement 1*). Data points represent the mean value of 15 consecutive EPSCs unless indicated and were plotted over time (e.g., *Figure 1B*).

Interneuron single action potentials (APs) or bursts of APs were evoked by either 15 ms or 700 ms depolarizing pulses (200–300 pA), respectively, that were applied 10 ms before the SC stimulation. Single APs and SC stimuli were paired for 3 min every 4 s. Protocol of pairing bursts of APs and SC stimuli was delivered 3 times every 2 s. To study the dynamic interplay between interneuron-astrocyte activity and excitatory synaptic transmission, bursts of APs were applied to interneurons by repetitive depolarizations (3 Hz, 30 s). SC stimuli were then phase-locked at either the interneuron depolarization (up-EPSC) or the resting state (down-EPSC; protocol shown in *Figure 5A*). In *Figure 5*, *Figure 5—figure supplements 1* and *2* each data point represents the simple moving average (*Jadhav et al., 2012*; *O'Connor et al., 2005*) of 15 consecutive EPSCs and were plotted over time. Bar graphs represent the mean value of the EPSCs at the 20–30 s periods after starting the protocol (e.g., *Figure 5B*).

Different depolarizing pulse durations were applied to the interneuron in order to achieve a broad range of AP firing. Synaptic efficacy values of the experiments were grouped according to the number of APs evoked in the interneuron (6-APs binning: 1–6, 7–12, 13–18, 19–24, and >24 APs) and the Hill equation was fitted to the data obtained (*Figure 1H*). The values of the fit ($R^2 = 0.9955$) were: minimum = 101.40% (95% confidence intervals [CI]: 91.56 to 111.3%), maximum = 159.60% (95% CI: 149.9 to 169.2%), IN firing activity at which the synaptic efficacy was 50% of the maximum = 13.00 APs (95% CI: 10.49 to 15.51 APs), and slope = 0.18 (95% CI: 0.02 to 0.34).

## Interneuron classification

A representative sample of interneurons ($n = 74$) were analyzed and classified according to their electrophysiological properties following the terminology of the 'Petilla interneuron nomenclature Group' (PinG) (*Ascoli et al., 2008*). Inter-spike interval (ISI) between each two consecutive spikes was calculated and then represented an 'ISI adaptation ratio' (ratio between each ISI divided by the first ISI ($ISI_n/ISI_{1st}$)) (*Ascoli et al., 2008*; *Kröner et al., 2007*). Three different populations of interneurons were identified: (1) Fast-spiking (FS) interneurons ($n = 25$; 33.8%), that exhibited a continuous firing pattern without frequency adaptation (ISI ratio <1.25); (2) Non-adapting, non-fast spiking (NAD/NFS) interneurons ($n = 29$; 39.2%), that exhibited an initial burst followed by a steady-state firing with no or minimal frequency adaptation; and (3) Adapting (AD) interneurons ($n = 20$; 27.0%), that showed a marked frequency adaptation (ISI ratio >2; *Figure 1—figure supplement 1D*). The potentiation of the CA3-CA1 excitatory synaptic transmission was independent of the type of interneuron stimulated (*Figure 1—figure supplement 1E*). In a subset of experiments parvoalbumin positive neurons (PV[+]) from PV-Cre knockin driver mice backcrossed to a Cre-responsive reporter line (Ai9-rcl-tdTomato transgenic mice were recorded (*Figure 1—figure supplement 2*).

## Calcium imaging

$Ca^{2+}$ levels in astrocytes were monitored by fluorescence microscopy using the $Ca^{2+}$ indicator Fluo-4-AM. Slices were incubated with Fluo-4-AM (2–5 μL of 2 mM dye were dropped over the hippocampus, attaining a final concentration of 2–10 μM and 0.01% of pluronic) for 20–30 min at room temperature. In order to confirm the specific recording of $Ca^{2+}$ signals in astrocytes, animals were injected intraperitoneally with sulforhodamine 101 (SR101; 100 mg/kg) 2 hr before sacrificed. In these conditions, astrocytes were specifically loaded with SR101 (e.g., *Figure 3A*) (*Nimmerjahn et al., 2004*; *Perez-Alvarez et al., 2014*). Additionally, astrocytes were confirmed by their electrophysiological properties (*Araque et al., 2002*; *Nimmerjahn et al., 2004*). Astrocytes were then imaged using a CCD camera (ORCA-235; Hamamatsu, Japan) attached to the microscope

(Olympus BX51WI). Cells were illuminated during 100–500 ms with a xenon lamp at 490 nm using a monochromator Polychrome V (TILL Photonics, Gräfelfing, Germany), and images were acquired every 0.5–1 s. The monochromator and the camera were controlled and synchronized by the IPLab software that was also used for quantitative epifluorescence measurements. Analysis of astrocyte $Ca^{2+}$ levels were restricted to the region of the cell body and $Ca^{2+}$ variations were estimated as changes in the fluorescence signal over the baseline ($\triangle F/F_0$). The astrocyte $Ca^{2+}$ signal was quantified from the probability of occurrence of a $Ca^{2+}$ elevation (termed as $Ca^{2+}$ transient), calculated as the number of $Ca^{2+}$ transient grouped in 5 s bins recorded from the astrocytes in the field of view (6–12 astrocytes per analyzed region) (*Navarrete and Araque, 2010*), and mean values were obtained by averaging different experiments. To test the effects of interneuron activity on $Ca^{2+}$-transient probability under different conditions, the respective mean basal (15 s before the stimulus) and maximum $Ca^{2+}$ transient probability (recorded 15 s after interneuron stimulation) from $\geq 5$ slices per condition were averaged and compared (e.g., *Figure 3B*). $Ca^{2+}$ responses from different slices were normalized calculating the '$Ca^{2+}$ transient probability index' as: [($Ca^{2+}$ transient probability after stimulus) − ($Ca^{2+}$ transient probability before stimulus)] / [($Ca^{2+}$ transient probability after stimulus) + ($Ca^{2+}$ transient probability before stimulus)] (e.g., *Figure 3C*).

In some experiments the genetically encoded $Ca^{2+}$ indicator GCaMP3 specifically expressed in astrocytes was used to monitor $Ca^{2+}$ signaling in the GB1-cKO mice by using confocal microscopy (Olympus FV300) (*Figures 6C,D*), and analyzed as described. Local application of ATP (10 mM) and baclofen (10 mM) were delivered by 5 s duration pressure pulses through a micropipette.

## Conditional, astrocyte-specific GABA_B receptor knockout mice (GB1-cKO mice)

Functional GABA_B receptor ablation was investigated in conditional, astrocyte-specific GABA_B receptor knockout mice (GB1-cKO), generated by crossbreeding *Gabbr*1^fl/fl (MGI:3512742) (*Haller et al., 2004*) with GLAST-CreERT2 knockin mice (MGI:3830051) (*Mori et al., 2006*). In some of the experiments, mice with astrocytes-specific expression of the genetically encoded $Ca^{2+}$ indicator GCaMP3 were used. For that purpose R26-lsl-GCaMP3 mice (JAX #014538) (*Paukert et al., 2014*) were crossbred to GB1-cKO and control mice. The selective deletion of the receptor subunit GABBR1 is sufficient to completely block functional GABA_B receptor activity (*Bettler et al., 2004*). To induce DNA recombination in GLAST-CreERT2xGABA_B^fl/fl or GLAST-CreERT2xR26-lsl-GCaMP3 mice (*Paukert et al., 2014*), tamoxifen (10 mg/ml corn oil, Sigma, St. Louis, USA) was intraperitoneally injected into 3-week-old mice on three consecutive days (100 mg/kg per body weight). 21 days after the first injection, mice were started to be analyzed. All mouse lines were maintained in the C57BL/6N background.

## Immunohistochemistry

The animals were anesthetized with Ketamine/Rompun (1.4% ketamine, 0.2 xylazin, 0.9% NaCl; 5 ml/kg per body weight) and intracardially perfused with ice cold ACFS and subsequently with 4% paraformaldehyde (PFA) in 0.1 M phosphate buffer (pH 7.4). The brain was removed, dissected into the two hemispheres, and post fixed for 4–6 hr in 4% PFA in 0.1 M phosphate buffer (pH 7.4) at 4°C. The fixed brain tissue was cut in phosphate buffered saline (PBS) into sagittal sections (50–70 μm thickness) at a Leica VT1000S vibrato (Leica, Nussloch, Germany). These sections were collected in 24-well tissue culture plates containing PBS. Vibratome sections were incubated for one hour in blocking buffer (0.3% Triton X-100, 5% horse serum in PBS) at RT (room temperature). The primary antibodies were diluted in the blocking solution and the sections were incubated overnight at 4°C. As marker for GABAergic interneurons mouse anti-GAD67 (RRID:AB_2278725; 1:500) was used. For astrocyte labeling the following antibodies were used: chicken anti-GFAP (RRID:AB_921444; 1:1000), and rabbit anti-GLAST (RRID:AB_304334; 1:250). GABA_B receptors were stained with a guinea pig anti-GABABr1 (RRID:AB_1587048; 1:500). The slices were washed three times for ten min each in 1xPBS. The secondary antibody was diluted in the secondary antibody buffer (2% horse serum in PBS) and incubated for 2 hr at room temperature. Secondary antibodies were donkey anti-mouse, anti-goat, and anti-rabbit (1:2000) conjugated with Alexa488, Alexa546, Alexa555, Alexa633, and purchased from Invitrogen (Thermo Fisher Scientific Inc). The sections were finally washed for 3 times with 1xPBS (10 min) and mounted in Aqua polymount (Polysciences). For astrocytic network labeling,

after biocytin filling slices were fixed in 4% PFA in 0.1 PBS (pH 7.4) at 4°C. Biocytin was visualized with Alexa488-Streptavidin (RRID:AB_2315383; 1:500) applied in the staining protocol described above for 48 hr.

## Microscopic analysis and quantification

Confocal images were recorded by laser scanning microscopy (LSM 710, Zeiss, Carl Zeiss AG, Jena) using a 40x objective (Plan-Aprochomat 40x/1, 4 Oil DIC (UV) VIS-IR M 27). For excitation of fluorescent dyes, a Lasos Argon laser (454 nm to 514 nm) and a Helium-Neon laser (543 nm, 633 nm) were used. Z-stacks of images were taken at 0.5 mm intervals and processed with ImageJ using the JACoP v2.0 colocalization plugin (*Cordelières and Bolte, 2014*). In brief, the deletion of *Gabbr1* was determined as a reduction of its immunolabel within the respective channel of the astroglial glutamate transporter GLAST. The overlap coefficient and Mander's coefficient M2 were determined. Although both coefficients have their unique limitations, both indicated a significant and astrocyte-specific reduction of *Gabbr1* and were plotted in *Figure 6B* (9 sections from 2 GB1-cKO and 11 sections from two control mice; 2-sided; unpaired *t* test). The analysis of co-localization probably underestimates the *Gabbr1* removal, since the spatial resolution in single optical sections is less than the size of the fine astrocyte processes contacting presynaptic terminals that are *Gabbr1*-positive as well.

## Quantitative real-time PCR (qRT-PCR)

Levels of messenger RNA (mRNA) and genomic DNA were detected by reverse transcriptase PCR. Hippocampi of 7 GB1-cKO and seven control mice (seven weeks old) were removed from the skull, homogenized (Precellys homogenizer, peqlab, Erlangen, Germany) and divided for RNA extraction with RNeasy mini kit (QIAGEN, Hilden, The Netherlands) as well as for DNA analysis RNA/DNA ALL Prep-Kit (QIAGEN, Hilden, The Netherlands). Successful gene recombination was determined by quantifying the loss of the loxP flanked gene region. Primers were located closely upstream and downstream of the 5' loxP site. Control and cKO were homozygously floxed for the *Gabbr1* locus (*Gabbr1*$^{fl/fl}$); controls were wild type for the GLAST locus (GLAST$^{+/+}$) and GB1-cKOs were heterozygous for the CreERT2 transgene in the GLAST locus (GLAST$^{CreERT2/+}$). Since only non-recombined alleles were amplified, reduction of the respective PCR signal indicated successful recombination. Values (△CT) of GB1-cKO animals were normalized to the mean △CT values of control animals.

For quantification of the PCR products, the fluorescent dye EvaGreen (Axon) was used. PCR runs were performed using the CFX96 Real-Time PCR Detection System (BioRad). All reactions were carried out in triplicates. Neuregulin one type III (NrgIII) and $\beta$-actin were used as endogenous gene controls. Data normalization and analysis were performed with the qbase+ real time PCR data analysis software (Biogazelle) based on the ΔΔCT-method.

Primer sequences for CDNA analysis by qRT-PCR were as follows (in 5' to 3' direction): ATPase forward GGA TCT GCT GGC CCC ATA C; ATPase reversed CTT TCC AAC GCC AGC ACC T, b-Actin forward CTT CCT CCC TGG AGA AGA GC; b-Actin reversed ATG CCA CAG GAT TCC ATA CC; *Gabbr1* forward CGA AGC ATT TCC AAC ATG AC; *Gabbr1* reversed CAA GGC CCA GAT AGC ATC ATA. Primer sequences for genomic DNA were as follows: NRGIII forward GTG TGC GGA GAA GGA GAA AAC T; NRGIII reversed AGG CAC AGA GAG GAA TTC ATT TCT TA; b-Actin forward CTG CTC TTT CCC AGA CGA GG; b-Actin reversed AAG GCC ACT TAT CAC CAG CC; *Gabbr1* forward CAG TCG ACA AGC TTA GTG GAT CC, *Gabbr1* reversed TCC TCG ACT GCA GAA TTC CTG.

## In vivo recordings

GB1-cKO mice and wild-type littermates (12–16 weeks) were placed in a stereotaxic frame under urethane anesthesia (1.8 g/kg, intraperitoneal injection), constantly monitored for body temperature and breathing rate, and kept warm with a heating pad. Electrodes were placed stereotaxically according to the atlas (*Paxinos and Franklin, 2012*). Local field potentials (LFP) were recorded through stainless steel macroelectrodes (1 MΩ) placed in the CA1 layer (AP, −2; L, 1.4; V, 1.1 mm from Bregma) and amplified (Differential AC Amplifier Model 1700, A-M System), bandpass filtered between 0.1 Hz and 500 kHz, and digitized at 100 kHz (PowerLab 4/25 T and LabChart, ADInstruments) running in a PC for direct visualization and storage. Then, two nichrome stimulating

electrodes (Isolated Pulse Stimulator Model 2100, A-M Systems) were placed in the vibrissae. After stabilization and basal activity recordings, an electrical stimulus (10 Hz, 10 s duration at 10 V) was applied to vibrissae. Three stimuli were applied with an interstimulus period of $\geq$5 min.

Six epochs (five second bins) during one minute in basal conditions were analysed. Also, the first 10 s, divided in 5 sec-bins, starting at the end of each stimulus were selected. Epoch was stored in a new file and converted to an adequate format to perform the spectral analyses (Clampfit 10.2, MDS Analytical Technologies). Spectral analyses for each bin were assessed by fast Fourier transformation through the Hamming window with 50% overlap, obtaining the power density ($V^2 \cdot Hz^{-1}$) with a spectral resolution of 0.38 Hz, from 0.38 to 100.3 Hz. Since animals had different levels of baseline power density, the power values for each frequency were normalized as a percentage of the total power density recorded before computing group results. After normalization six epochs were averaged for basal and post-stimuli condition and compared between control littermate and GB1-cKO mice. We selected the following frequency bands: theta, 4–8 Hz; low gamma, 30–50 Hz and high gamma, 70–90 Hz.

For *phase-amplitude* coupling (PAC) analysis each 5 s bin was converted to text format to perform the computation trough MATLAB (The MathWorks, Inc.). The process was performed by a custom-made script on MATLAB (https://github.com/abdel84/). Raw signal was decimated to a sample rate of 1 kHz, then an elliptical filter was applied to remove frequencies below 3 Hz and two additional bandpass filters for both Theta (4–8 Hz) and Gamma (30–80 Hz) bands. The Theta phase and the Gamma amplitude, respectively, were extracted and computed to obtain their time series using the standard Hilbert transform as described previously (*Tort et al., 2010*) and to obtain the Phase-Locking Value (PLV). This index represents the degree to which the Gamma amplitude is comodulated with the Theta phase and ranges between 0 and 1, with higher values indicating stronger PAC interactions (*Tort et al., 2010*). To calculate the mean vector of PLV, circular statistics analysis was performed by using CircStat toolbox[19] and then normalized by Fisher's Z Transformation, to apply regular statistical analysis: z' = 0.5 [ln (1+r) − ln (1− r)].

## Drugs and chemicals

*N*-(Piperidin-1-yl)−5-(4-iodophenyl)−1-(2,4-dichlorophenyl)−4-methyl-1*H*-pyrazole-3-carboxamide (AM251), (*S*)-(+)-α-Amino-4-carboxy-2-methylbenzeneacetic acid (LY-367385), 2-Methyl-6-(phenylethynyl)pyridine hydrochloride (MPEP), (2*S*)−3-[[(1*S*)−1-(3,4-Dichlorophenyl)ethyl]amino-2-hydroxypropyl](phenylmethyl)phosphinic acid hydrochloride (CGP55845), and (*R*)-Baclofen were purchased from Tocris (Bristol, UK). The $Ca^{2+}$ indicator Fluo-4-AM was purchased from Life Technologies Ltd (Paisley, UK). Picrotoxin, atropine, thapsigargin, and 1,2-bis(2-aminophenoxy)ethane-*N*,*N*,*N′*,*N′*-tetraacetate (BAPTA) were purchased from Sigma-Aldrich (St. Louis, MO, USA).

## Statistical analysis

The normality test was performed before applying statistical comparisons, which were made using non parametric Wilcoxon Rank-sum Test and parametric Student's *t* tests as deemed appropriate. Two-tailed, unpaired or paired *t* test was used for comparisons unless indicated. Data are expressed as mean ± standard error of the mean (SEM). When a statistical test was used, the precise two-sided *P* value and the test employed are reported in the text and/or figure legends. Statistical differences were established with p<0.05 (*), p<0.01 (**), and p<0.001 (***, #). Blind experiments were not performed in the study but the same criteria were applied to all allocated groups for comparisons. Randomization was not employed. The sample size in whole-cell recording experiments was based on the values previously found sufficient to detect significant changes in hippocampal synaptic strength in past studies from the lab. For in vivo recordings an N of 3 repetitions of stimuli were applied, and independent recordings were summarized from six animals per condition, which provided sufficient statistical power while trying to minimize the number of animals sacrificed.

## Acknowledgements

Authors thank Dr J Chen (UCSD, CA, USA) for providing *Ip3r2*$^{-/-}$ mice, Dr M Götz (Helmholtz Center Munich, Germany) for GLAST-CreERT2 knockin mice, Drs W Buño, M Navarrete, and R Martín for comments, and B Pro for technical assistance. This work was supported by MINECO (Consolider, CSD2010-00045; Ramón y Cajal Program RYC-2012–12014; BFU2013-47265R; and BFU2016-75107-

P) to GP; Juan de la Cierva Program (MINECO, JCI-2011–09144 and IJCI-2014–19136) to RG; International Graduate School of Neuroscience (IGSN. FNO 01/114) to AR; FONDECYT 1130614 and Millennium Nucleus NUMIND (NC-130011) to MF; German Research Foundation (SFB 874/B1) to DM-V; MINECO (BFU2011-26339), INCRECyT project from European Social Fund, PCyTA and JCCM to EDM; Swiss National Science Foundation (3100 A0-117816) to BB; DFG SPP 1757, DFG SFB 894, EC FP7-People ITN-237956 EdU-Glia to FK; EC FP7-Health-202167 NeuroGLIA to FK and AA Cajal Blue Brain, MINECO (BFU2010-15832), Human Frontier Science Program (RGP0036/2014), and NIH-NINDS (R01NS097312-01) to AA.

## Additional information

### Funding

| Funder | Grant reference number | Author |
|---|---|---|
| Ministerio de Economía y Competitividad | Consolider, CSD2010-00045 | Gertrudis Perea |
| Ministerio de Economía y Competitividad | Ramón y Cajal Program, RYC-2012–12014 | Gertrudis Perea |
| Ministerio de Economía y Competitividad | BFU2013-47265R | Gertrudis Perea |
| Ministerio de Economía y Competitividad | BFU2016-75107-P | Gertrudis Perea |
| Ministerio de Economía y Competitividad | Juan de la Cierva Program, JCI-2011–09144 | Ricardo Gómez |
| Ministerio de Economía y Competitividad | IJCI-2014-19136 | Ricardo Gómez |
| International Graduate School of Neuroscience | IGSN. FNO 01/114 | Abdelrahman Rayan |
| Fondo Nacional de Desarrollo Científico y Tecnológico | 1130614 | Marco Fuenzalida |
| Millennium Nucleus NUMIND | NC-130011 | Marco Fuenzalida |
| Schweizerischer Nationalfonds zur Förderung der Wissenschaftlichen Forschung | 3100 A0-117816 | Bernhard Bettler |
| Deutsche Forschungsgemeinschaft | SFB 874/B1 | Denise Manahan-Vaughan |
| Ministerio de Economía y Competitividad | BFU2011-26339 | Eduardo D Martín |
| European Social Fund | INCRECyT project | Eduardo D Martín |
| European Commission | FP7-People ITN-237956, EdU-Glia | Frank Kirchhoff |
| Deutsche Forschungsgemeinschaft | SPP 1757 | Frank Kirchhoff |
| Deutsche Forschungsgemeinschaft | SFB 894 | Frank Kirchhoff |
| Human Frontier Science Program | RGP0036/2014 | Alfonso Araque |
| National Institute of Neurological Disorders and Stroke | NIH-NINDS (R01NS097312-01) | Alfonso Araque |

The funders had no role in study design, data collection and interpretation, or the decision to submit the work for publication.

## Author ORCIDs

Gertrudis Perea, http://orcid.org/0000-0001-5924-9175

Amit Agarwal, http://orcid.org/0000-0001-7948-4498

## Ethics

Animal experimentation: All the procedures for handling and sacrificing animals followed the European Commission guidelines for the welfare of experimental animals (2010/63/EU), US National Institutes of Health and the Institutional Animal Care and Use Committee at the University of Minnesota (USA). The use of astrocyte-specific GABBR1 knockout mice was approved by the Saarland state´s "Landesamt für Gesundheit und Verbraucherschutz" in Saarbrücken/Germany (animal license number 72/2010).

## Author contributions

GP, Designed the study, Performed experiments, Analyzed data and wrote the paper, Conception and design, Acquisition of data, Analysis and interpretation of data, Drafting or revising the article; RG, Designed the study, Performed experiments, Analyzed data, Conception and design, Acquisition of data, Analysis and interpretation of data, Drafting or revising the article ; SM, AC, JJB, AH-V, MM-F, RQ, AD, Acquisition of data, Analysis and interpretation of data, Drafting or revising the article; LS, Acquisition of data, Analysis and interpretation of data; AR, Analysis and interpretation of data, Drafting or revising the article; MF, Acquisition of data, Analysis and interpretation of data, Drafting or revising the article; AAg, Provided the astrocyte-GCaMP3 mouse line, Drafting or revising the article, Contributed unpublished essential data or reagents; DEB, Provided the astrocyte-GCaMP3 mouse line, Drafting or revising the article, Contributed unpublished essential data or reagents; BB, Provided the GABBR1fl/fl mouse line, Drafting or revising the article, Contributed unpublished essential data or reagents; DM-V, Analysis and interpretation of data,Drafting or revising the article; EDM, Conception and design, Acquisition of data, Analysis and interpretation of data, Drafting or revising the article; FK, Acquisition of data, Analysis and interpretation of data, Drafting or revising the article, Contributed unpublished essential data or reagents; AAr, Designed the study and contributed significantly to the interpretation and revising and writing the article, Conception and design, Analysis and interpretation of data, Drafting or revising the article

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
