## [Decision Letter]

Thank you for submitting your article "Activity-dependent switch of GABAergic inhibition into glutamatergic excitation in astrocyte-neuron networks" for consideration by *eLife*. Your article has been reviewed by two peer reviewers, one of whom is a member of our Board of Reviewing Editors, and the evaluation has been overseen by Gary Westbrook as the Senior Editor.

The reviewers have opted to remain anonymous. The reviewers have discussed the reviews with one another and the Reviewing Editor has drafted this decision to help you prepare a revised submission.

Summary:

This study examines the interaction between interneurons and astrocytes and their impact on glutamatergic transmission of Schaffer collateral synapses targeting CA1 principal cells. Single action potentials induced in GABAergic cells reduced synaptic efficacy and success rate of synaptic transmission whereas bursts of action potentials in GABAergic cells induced the opposite, a potentiation of synaptic efficacy and success rate of transmission. The first depressing effect was due to presynaptic GABAA receptor activation at Schaffer collateral synapses whereas the potentiation was mediated by GABAB receptors expressed at astrocytes. Activation of GABABRs increases intracellular Ca^2+^ signals and thereby induced glutamate release, which in turn presynaptically enhances glutamate release by acting at presynaptic metabotropic group I glutamatergic receptors. This is an elegant study providing intriguing and provocative data that identify astrocytic glutamate release as an interesting component of the GABAergic modulation of neuronal excitatory activity.

Essential revisions:

1) The results suggest that some synapses express both GABAA and GABAB receptors. Recent studies have described that the expression of GABA receptors can be altered in pathological conditions (i.e. during hypoxic conditions; Zonouzi et al., Nature Neurosci. 2015). It is difficult to judge whether such GABA receptor expression represent a "physiological condition" or rather a result of "traumatic injury and hypoxia" (Takano et al., Glia 2014). Please include control experiments to potentially exclude such effects or alternatively, if hypoxia cannot be excluded, add a Discussion section to address this issue.

2) Please test in these experiments whether only glutamate or also other amino acid transmitters such as D-serine can be released by astrocytes.

3) Please consider using the terminology Ca^2+^ signal or transient instead of Ca^2+^ spike. The Results section also requires additional information on how cells were loaded and which indicators have been applied. The authors point to the probability of obtaining Ca^2+^ signals. Information on changes in the shape (amplitude, time course) of Ca^2+^ signals over time would be important to gain insights on the temporal changes and stability in the signal upon repetitive induction (see Figure 3).

4) The schematic in Figure 4 is very helpful but the inclusion of the functional outcome (potentiation vs depression of synaptic efficacy) should be included.

5) It is very hard to identify co-localization of the various markers in Figure 6. Please include magnified insets in the figure to allow a better inspection of the co-localization of GABABR1 in controls vs KOs in astrocytes.

6) Differences in the normalized power spectrum of network oscillations in controls and GABABR1 KOs in Figure 7 is small. This should be emphasized in the Results section so as not to overinterpret the results. Due to the mild differences it is unclear why the relative differences of theta and γ power in Figure 7 are so large. The power values were determined in relation to what?

7) The astrocyte specific cKO of GABBR1 in Figure 6 is a critical component of the authors' overall hypothesis. However the data in Figure 6 are uninterpretable at least at the magnification shown. Also the qPCR data of the fl/fl allele in hippocampus cannot be used to assess the completeness of the KO. Although the lack of a Ca response in the cKO astrocytes is used as control for the GABABR1 cKO, the calcium imaging was performed in astrocyte somata yet it is known that, e.g. in the IP3R2 KO, there can ongoing calcium signals in astrocyte processes in the absence of Ca signals in the soma. The authors should discuss how this might affect their results, given that one might expect that astrocyte-synaptic interactions would occur primarily at astrocyte processes.

---

## [Author Response]

*Essential revisions:*

*1) The results suggest that some synapses express both GABAA and GABAB receptors. Recent studies have described that the expression of GABA receptors can be altered in pathological conditions (i.e. during hypoxic conditions; Zonouzi et al., Nature Neurosci. 2015). It is difficult to judge whether such GABA receptor expression represent a "physiological condition" or rather a result of "traumatic injury and hypoxia" (Takano et al., Glia 2014). Please include control experiments to potentially exclude such effects or alternatively, if hypoxia cannot be excluded, add a Discussion section to address this issue.*

Following the reviewer’s suggestion the following paragraph has been included to discuss this issue: “Some evidence have shown that acute brain slices might undergo hypoxic conditions causing reactive changes in astrocytes (Takano et al., 2014), and a downregulation of GABAA receptor expression (Zonouzi et al., 2015); however, since the study of molecular individualities of the interneuron-astrocyte signaling show limitations that need to be explored ex vivo, these associated alterations and their potential influence cannot not be excluded from the observed responses.”

*2) Please test in these experiments whether only glutamate or also other amino acid transmitters such as D-serine can be released by astrocytes.*

We acknowledge the reviewer´s suggestion and a new set of experiments to evaluate D-serine contribution were performed. After assessing that synapses experienced potentiation by interneuron stimulation, the antagonist of NMDA receptors AP5 (50 mM) was perfused. In presence of AP5 synaptic strength was still enhanced by interneuron stimulation, suggesting that putative D-serine released by astrocytes and NMDA receptors activation were not responsible for the observed effects.

These data have been included in the new Figure 4 and in the text as follows: “Additionally, recent studies have reported the contribution of D-serine released by astrocytes to synaptic plasticity through NMDA receptors activation (Henneberger et al., 2010; Takata et al., 2011). However, the synaptic potentiation induced by interneuron activity was unaffected by the perfusion of the NMDA receptor antagonist AP5 (50 µM; *n* = 6; *P* = 0.11; paired *t* test) (Figure 4); suggesting that EPSC modulation was independent of the astrocytic D-serine actions.”

*3) Please consider using the terminology Ca^2+^ signal or transient instead of Ca^2+^ spike. The Results section also requires additional information on how cells were loaded and which indicators have been applied. The authors point to the probability of obtaining Ca^2+^ signals. Information on changes in the shape (amplitude, time course) of Ca^2+^ signals over time would be important to gain insights on the temporal changes and stability in the signal upon repetitive induction (see Figure 3).*

Following the reviewer´s suggestion “Ca^2+^ spike” terminology was changed for “Ca^2+^ transient”. The information regarding the loading protocols and indicators has been revised accordingly in the methods section.

Regarding Figure 3, it shows representative somatic Ca responses from 4 different astrocytes evoked by consecutive interneuron stimulation with one single and burst of action potentials. As we indicated in the previous version, to test the effects of interneuron activity on Ca^2+^ signals, the presence of Ca events happening at the soma were evaluated before and after interneuron stimulation.

Present data show that both the pharmacological blockage of GABAB receptors and the genetic downregulation of *Gabab1* expression strongly reduced the interneuron-evoked Ca^2+^ transients in astrocyte cell body and impaired the EPSC potentiation induced by interneuron activity (Figure 1,Figure 3,Figure 6), suggesting a crucial role of the Ca^2+^ signals coming from the astrocytic soma in the observed synaptic modulation. Therefore, although we agree with the referee that a detailed analysis of the temporal Ca^2+^ changes would bring more information about the GABAergic impact on astrocyte Ca^2+^ signaling, which will be exhaustively examined in future studies; we consider that such analysis does not compromise either the current data or the main message of the manuscript.

*4) The schematic in Figure 4 is very helpful but the inclusion of the functional outcome (potentiation vs depression of synaptic efficacy) should be included.*

Figure 4 has been revised accordingly to clarify functional outcome of this signaling.

*5) It is very hard to identify co-localization of the various markers in Figure 6. Please include magnified insets in the figure to allow a better inspection of the co-localization of GABABR1 in controls vs KOs in astrocytes.*

Following the reviewer’s suggestion, the new Figure 6 includes new panels to more clearly show co-localization of GABABR1 in controls vs KOs in astrocytes.

*6) Differences in the normalized power spectrum of network oscillations in controls and GABABR1 KOs in Figure 7 is small. This should be emphasized in the Results section so as not to overinterpret the results. Due to the mild differences it is unclear why the relative differences of theta and γ power in Figure 7 are so large. The power values were determined in relation to what?*

Following the reviewer’s comment we have revised the Figure 7 in order to clearly show the differences between the frequency bands of interest. The new Figure 7 (insets) shows the mean values for each frequency range (previous Figure 7 represented the area mean values for the corresponding frequency bands).

The difference between controls and GABABR1 KOs has been emphasized accordingly in Results and Discussion section as follows:

“Both theta band (peak 4-8 Hz; P = 0.007; unpaired t test) and low γ band activities (peak 30-50 Hz; P = 0.042; Wilcoxon rank-sum test) were partially reduced in GB1-cKO mice (Figure 7).”

“Present data from astrocyte *Gabbr1* knockout mice show a partial but significant decrease of stimulus-induced theta-γ oscillations and coupling.”

To clarify the normalization of power values the following sentence has been included in the text: “Since animals had different levels of baseline power density, the power values for each frequency were normalized as a percentage of the total power density recorded before computing group results. After normalization six epochs were averaged for basal and post-stimuli condition and compared between control littermate and GB1-cKO mice. We selected the following frequency bands: theta, 4–8 Hz; low γ, 30–50 Hz and high γ, 70-90 Hz.”

*7) The astrocyte specific cKO of GABBR1 in Figure 6 is a critical component of the authors' overall hypothesis. However the data in Figure 6 are uninterpretable at least at the magnification shown. Also the qPCR data of the fl/fl allele in hippocampus cannot be used to assess the completeness of the KO. Although the lack of a Ca response in the cKO astrocytes is used as control for the GABABR1 cKO, the calcium imaging was performed in astrocyte somata yet it is known that, e.g. in the IP3R2 KO, there can ongoing calcium signals in astrocyte processes in the absence of Ca signals in the soma. The authors should discuss how this might affect their results, given that one might expect that astrocyte-synaptic interactions would occur primarily at astrocyte processes.*

Following the reviewer’s comments, the new Figure 6 includes new panels to more clearly show co-localization of GABABR1 in controls vs KOs in astrocytes (comment #5).

Additionally, a new paragraph has been included to discuss the Ca^2+^ signaling at the processes: “In addition, present data cannot discard that residual Ca^2+^ events might occur in the fine process of the GB1-cKO astrocytes, as they have been found for *Ip3r2^-/-^* mice (Srinivasan et al., 2015); however, the existence of those events would not be sufficient to induce the synaptic potentiation observed after interneuron stimulation (Figure 3; Figure 6). Thus, these data suggest that although the astrocyte-synaptic interactions might primarily take place at the astrocyte processes, the synaptic plasticity induced by interneuron-astrocyte communication is a highly regulated phenomenon that requires the active contribution of astrocyte somatic Ca^2+^ signaling.”